# Optimal Stochastic and Online Learning with Individual Iterates

**Yunwen Lei**[1,2]   **Peng Yang**[1]   **Ke Tang**[1*]   **Ding-Xuan Zhou**[3]

[1]University Key Laboratory of Evolving Intelligent Systems of Guangdong Province,
Department of Computer Science and Engineering,
Southern University of Science and Technology, Shenzhen 518055, China

[2]Department of Computer Science,
Technical University of Kaiserslautern, Kaiserslautern 67653, Germany

[3]School of Data Science and Department of Mathematics,
City University of Hong Kong, Kowloon, Hong Kong, China

`{leiyw, yangp, tangk3}@sustech.edu.cn`   `mazhou@cityu.edu.hk`

## Abstract

Stochastic composite mirror descent (SCMD) is a simple and efficient method able to capture both geometric and composite structures of optimization problems in machine learning. Existing strategies require to take either an average or a random selection of iterates to achieve optimal convergence rates, which, however, can either destroy the sparsity of solutions or slow down the practical training speed. In this paper, we propose a theoretically sound strategy to select an *individual* iterate of the vanilla SCMD, which is able to achieve optimal rates for both convex and strongly convex problems in a non-smooth learning setting. This strategy of out-putting an individual iterate can preserve the sparsity of solutions which is crucial for a proper interpretation in sparse learning problems. We report experimental comparisons with several baseline methods to show the effectiveness of our method in achieving a fast training speed as well as in outputting sparse solutions.

## 1 Introduction

Gradient-based methods have found wide applications to solve various optimization problems. A basic and representative method of this type is the gradient descent, which iteratively moves iterates along the minus gradient direction of the current iterate. However, gradient descent applied to machine learning problems requires to go through all training examples at each iteration, which is not efficient when the sample size is large. Stochastic gradient descent (SGD) relieves this computational burden by approximating the true gradient of the objective function with an unbiased estimation based on a randomly selected training example. With this strategy, SGD can achieve sample-size independent computational cost per iteration and therefore can be successfully applied to very large datasets which are becoming ubiquitous in the big data era [2, 4, 44].

From different viewpoints, SGD has been extended in various ways. For example, the trick of variance-reduction has been introduced to exploit the finite summation structure of objective functions for reducing the inherent variance [16, 33, 40, 43]. Adaptive step sizes were proposed to dynamically incorporate the knowledge of the geometry of the data observed [9]. Decreasing step sizes in a stagewise manner is used as a common trick in practice [14, 19, 41]. The trick of momentum is widely used to accelerate SGD by choosing an appropriate direction to pursue per iteration [18, 26, 29]. Proximal operators have been introduced to capture a structure of optimization problems [28], which

---

separates the regularizer and data-fitting terms to achieve a desired regularization effect. The concept of mirror map has been introduced to induce a Bregman distance for reflecting the geometry of the associated optimization problem [3, 25]. Empirical studies have shown that these variants can further improve the practical performance of SGD [4]. In this paper, we consider stochastic composite mirror descent (SCMD), which combines the technique of mirror map and proximal operator to capture both the composite and geometry structure of the associated optimization problem.

Theoretical studies of SCMD have attracted much attention since its appearance and many results have been derived to understand its promising behavior in practice. If the objective function is convex, then the expected suboptimality of the uniform average of iterates decays with the rate $O(T^{-\frac{1}{2}})$ after $T$ iterations [6–8, 30, 46]. If the objective function is strongly convex, then the expected suboptimality of the uniform average of iterates decays with the rate $O(T^{-1} \log T)$ [17, 35], which is unimprovable if the objective function is not smooth [31]. Surprisingly, some averaging schemes have been proposed to achieve the optimal rate $O(T^{-1})$ [14, 20, 31, 37]. Although taking averages of previous iterates can achieve theoretically minimax optimal rates [1], it can have some practical side effects [31, 36, 37][2]. For example, it can destroy the sparsity of the solution which is often crucial for a proper interpretation of models in many applications. Moreover, averaging can affect the practical training speed due to the existence of some possibly poor iterates in the iterate sequence produced by SCMD [31, 36]. The side effect of destroying sparsity can be resolved by randomly drawing an individual iterate with probabilities proportional to the weights in the optimal averaging scheme. However, this strategy of selecting a random iterate would introduces new variances, which may further slow down the training speed due to the possibility of selecting a poor iterate. Outputting the last SGD iterate is a natural strategy in practice, which would not affect sparsity and practical training speed. However, the suboptimality for the last SGD iterate converges with suboptimal rates $O(T^{-\frac{1}{2}} \log T)$ and $O(T^{-1} \log T)$ in the convex and strongly convex cases [37], respectively. Very recently, it was shown that the last SGD iterate can attain at most a rate $O(T^{-1} \log T)$ with high probabilities [13]. These facts motivate us to ask a natural question: can we develop an algorithm which can inherit the optimal convergence rate from the averaging strategies as well as the sparsity-preservation and fast practical training speed from the last SGD iterate?

In this paper, we aim to give an affirmative answer to the above question by proposing a novel strategy to select an individual iterate able to achieve theoretically optimal rates and fast training speed in practice. We first consider the case with the number of iterations known to us, which allows us to divide the implementation into two stages. The first stage produces an output with some averaging scheme, which is then used in the second stage to select an individual iterate according to the one-step progress evolution of SCMD. We then extend this algorithm to the online learning setting where training examples arrive in a sequential way. We show that our algorithm can achieve the optimal rates $O(T^{-\frac{1}{2}})$ and $O(T^{-1})$ in the convex and strongly convex case, respectively. Rooted on a careful one-step progress analysis, our selection of iterates is based on the difference of two successive iterates, which shares some spirits with the common heuristic trick of terminating an algorithm if the distance of two successive iterates is lower than a threshold. Our analysis also removes bounded subgradient assumptions in the existing discussions of optimal learning rates [14, 20, 31, 37]. We report experimental results to confirm the effectiveness of the proposed algorithm in both attaining a fast training speed and producing a sparse solution.

We present algorithms with motivation in Section 2. Theoretical results and discussions are given in Sections 3 and 4. Experimental results and conclusions are presented in Sections 5 and 6.

## 2   Algorithms with Motivations

### 2.1   Background

In supervised learning, we aim to infer an unknown relationship between input and output variables from a sequence of training examples $\{z_t = (x_t, y_t)\}_{t \in \mathbb{N}}$ drawn from a probability measure defined in a sample space $\mathcal{Z} = \mathcal{X} \times \mathcal{Y}$ with an input space $\mathcal{X} \subset \mathbb{R}^d$ and an output space $\mathcal{Y} \subset \mathbb{R}$, where $d \in \mathbb{N}$ is the dimension. A very elementary and powerful approximation of this relationship is a linear model of the form $x \mapsto \langle \mathbf{w}, x \rangle$ with $\mathbf{w} \in \mathbb{R}^d$, whose behavior at a single training example $(x, y)$ can

be quantified by a function $f(\mathbf{w}, z) = \ell(\langle \mathbf{w}, x \rangle, y)$, where the loss function $\ell : \mathbb{R}^2 \mapsto \mathbb{R}_+$ is convex with respect to (w.r.t.) the first argument and $\langle \mathbf{w}, x \rangle$ denotes the inner product between $\mathbf{w}$ and $x$. The learning process can be often formulated as an optimization problem of a composite structure

$$\min_{\mathbf{w} \in \mathbb{R}^d} \phi(\mathbf{w}) = \mathbb{E}_z[f(\mathbf{w}, z)] + r(\mathbf{w}), \tag{2.1}$$

where $\mathbb{E}_z[\cdot]$ denotes the expectation w.r.t. $z$ and $r : \mathbb{R}^d \mapsto \mathbb{R}_+$ is a regularizer possibly inducing sparsity. With specific instantiations of $\ell$ and $r$, Eq. (2.1) covers many famous learning problems in a unifying framework, including least squares, SVMs, logistic regression, lasso and elastic-net [8, 39].

SCMD provides an efficient first-order method to exploit the composite and geometry structure of the problem (2.1) [8, 24]. It extends SGD by using a strongly convex and differentiable mirror map $\Psi$ to generate an appropriate Bregman distance [3, 25] $D_\Psi(\mathbf{w}, \tilde{\mathbf{w}}) = \Psi(\mathbf{w}) - \Psi(\tilde{\mathbf{w}}) - \langle \mathbf{w} - \tilde{\mathbf{w}}, \nabla \Psi(\tilde{\mathbf{w}}) \rangle$, where $\nabla \Psi(\tilde{\mathbf{w}})$ denotes the gradient of $\Psi$ at $\tilde{\mathbf{w}}$. Let $\mathbf{w}_1 \in \mathbb{R}^d$ be an initial point and $\{\eta_t\}_{t \in \mathbb{N}}$ be a step size sequence. Up on the arrival of $z_t$ at the $t$-th iteration, we calculate a subgradient $f'(\mathbf{w}_t, z_t) \in \partial_w f(\mathbf{w}_t, z_t)$ as an unbiased estimate of $F'(\mathbf{w}_t) \in \partial F(\mathbf{w}_t)$, where $\partial_w f(\mathbf{w}_t, z_t)$ denotes the subdifferential of $f(\cdot, z_t)$ at $\mathbf{w}_t$ and $F(\mathbf{w}) = \mathbb{E}_z[f(\mathbf{w}, z)]$. SCMD updates the model by

$$\mathbf{w}_{t+1} = \arg\min_{\mathbf{w} \in \mathbb{R}^d} D_\Psi(\mathbf{w}, \mathbf{w}_t) + \eta_t \big( \langle \mathbf{w} - \mathbf{w}_t, f'(\mathbf{w}_t, z_t) \rangle + r(\mathbf{w}) \big). \tag{2.2}$$

Intuitively, SCMD uses $f'(\mathbf{w}_t, z_t)$ to form a first-order approximation of $f(\cdot, z_t)$ at $\mathbf{w}_t$ and uses the Bregman distance $D_\Psi(\mathbf{w}, \mathbf{w}_t)$ to keep $\mathbf{w}_{t+1}$ not far away from the current iterate. The regularizer $r$ is kept intact here to preserve a regularization effect. Typical choices of mirror maps include the $p$-norm divergence $\Psi_p(\mathbf{w}) = \frac{1}{2}\|\mathbf{w}\|_p^2$ ($1 < p \leq 2$), which works favorably when the solution of (2.1) is sparse by setting $p$ close to 1 [8, 39]. Here $\|\cdot\|_p$ is the $p$-norm defined by $\|\mathbf{w}\|_p = \big( \sum_{i=1}^d |w(i)|^p \big)^{1/p}$ for $\mathbf{w} = (w(1), \ldots, w(d))^\top \in \mathbb{R}^d$. SCMD recovers the stochastic proximal gradient descent by taking $\Psi = \Psi_2$ [7] and stochastic mirror descent by taking $r(\mathbf{w}) = 0$ [14]. It should be mentioned that $\rho$ is not necessarily to be an empirical distribution over training examples, and therefore the setting we consider here is more general than stochastic learning to minimize an empirical risk.

---

**Algorithm 1:** SCMDI

**Input:** $\{\eta_t\}_t, \sigma_\phi, \mathbf{w}_1$ and $T$.
**Output:** an approximate solution of (2.1)
1 **if** $\sigma_\phi == 0$ **then**
2 $\quad$ $s_w \leftarrow \mathbf{w}_1, \ s \leftarrow 1$
3 **else**
4 $\quad$ $s_w \leftarrow 6\eta_1 \mathbf{w}_1, \ s \leftarrow 6\eta_1$

5 **for** $t = 1, 2$ **to** $T - 1$ **do**
6 $\quad$ calculate $\mathbf{w}_{t+1}$ by (2.2)
7 $\quad$ **if** $\sigma_\phi == 0$ **then**
8 $\quad\quad$ $s_w \leftarrow s_w + \mathbf{w}_{t+1}, \ s \leftarrow s + 1$
9 $\quad$ **else**
10 $\quad\quad$ $s_w \leftarrow s_w + (t+2)(t+3)\eta_{t+1}\mathbf{w}_{t+1}$
11 $\quad\quad$ $s \leftarrow s + (t+2)(t+3)\eta_{t+1}$

12 $\bar{\mathbf{w}}_T \leftarrow s_w / s$
13 **for** $t = T, T+1$ **to** $2T - 1$ **do**
14 $\quad$ calculate $\mathbf{w}_{t+1}$ by (2.2)
15 $\quad$ $\triangle \leftarrow D_\Psi(\bar{\mathbf{w}}_T, \mathbf{w}_t) - D_\Psi(\bar{\mathbf{w}}_T, \mathbf{w}_{t+1})$
16 $\quad$ **if** $\triangle \leq T^{-1} D_\Psi(\bar{\mathbf{w}}_T, \mathbf{w}_T)$ **then**
17 $\quad\quad$ $T^* \leftarrow t, \ \mathbf{w}_{T^*} \leftarrow \mathbf{w}_t$

18 **return** $\mathbf{w}_{T^*}$

---

**Algorithm 2:** OCMDI

**Input:** $\{\eta_t\}_t, \sigma_\phi$ and $\mathbf{w}_1$.
**Output:** an approximate solution of (2.1)
1 **if** $\sigma_\phi == 0$ **then**
2 $\quad$ $s_w \leftarrow \mathbf{w}_1, \ s \leftarrow 1$
3 **else**
4 $\quad$ $s_w \leftarrow 6\eta_1 \mathbf{w}_1, \ s \leftarrow 6\eta_1$

5 $\bar{\mathbf{w}} \leftarrow \mathbf{w}_1, \ \hat{\mathbf{w}} \leftarrow \mathbf{w}_1, \ k \leftarrow 1$

6 **for** $t = 1, 2, \cdots$ **do**
7 $\quad$ calculate $\mathbf{w}_{t+1}$ by (2.2)
8 $\quad$ $\triangle \leftarrow D_\Psi(\bar{\mathbf{w}}, \mathbf{w}_t) - D_\Psi(\bar{\mathbf{w}}, \mathbf{w}_{t+1})$
9 $\quad$ **if** $\triangle \leq 2^{1-k} D_\Psi(\bar{\mathbf{w}}, \hat{\mathbf{w}})$ **then**
10 $\quad\quad$ $\tilde{\mathbf{w}} \leftarrow \mathbf{w}_t$
11 $\quad$ **if** $\sigma_\phi == 0$ **then**
12 $\quad\quad$ $s_w \leftarrow s_w + \mathbf{w}_{t+1}, \ s \leftarrow s + 1$
13 $\quad$ **else**
14 $\quad\quad$ $s_w \leftarrow s_w + (t+2)(t+3)\eta_{t+1}\mathbf{w}_{t+1}$
15 $\quad\quad$ $s \leftarrow s + (t+2)(t+3)\eta_{t+1}$
16 $\quad$ **if** $t == 2^k - 1$ **then**
17 $\quad\quad$ $k \leftarrow k + 1, \ \bar{\mathbf{w}} \leftarrow s_w / s, \ \hat{\mathbf{w}} \leftarrow \mathbf{w}_t$

18 **return** $\tilde{\mathbf{w}}$

---

## 2.2 Algorithms

We now present our first algorithm (Algorithm 1) which performs SCMD in two stages. In the first stage, we simply perform updates according to (2.2). An output $\bar{\mathbf{w}}_T$ is then produced by taking some

averages of the previous iterates. It is clear that $\bar{\mathbf{w}}_T$ is a simple average with uniform weights in the convex case, while the construction of $\bar{\mathbf{w}}_T$ in the strongly convex case is motivated by theoretical analysis (see Theorem 4) [20]. In the second stage, other than updating $\{\mathbf{w}_t\}$ with (2.2), we also search a time index (line 17 of Algorithm 1) at which the difference between two consecutive Bregman distance is less than a small number. We refer to Algorithm 1 as Stochastic Composite Mirror Descent with Individual iterates (SCMDI) since it outputs an individual iterate of the vanilla SCMD.

**Remark 1.** For any $t$, denote $A_t := D_\Psi(\bar{\mathbf{w}}_T, \mathbf{w}_t) - D_\Psi(\bar{\mathbf{w}}_T, \mathbf{w}_{t+1})$. It should be mentioned that the function $t \mapsto A_t$ is not monotonically w.r.t. $t$. Therefore, not all $t \geq T^*$ can necessarily satisfy the condition in line 16 of Algorithm 1, which is a requirement to achieve optimal convergence rates in our analysis.

In Algorithm 1, we update $T^*$ once we find a new $t$ satisfying $A_t \leq T^{-1}D_\Psi(\bar{\mathbf{w}}_T, \mathbf{w}_T)$. Another feasible strategy is to set $T^* = \arg\min_{t \in \{T, T+1, \ldots, 2T-1\}} A_t$. However, experiments show that this strategy does not work as well as Algorithm 1.

Algorithm 2 is an extension of Algorithm 1 by removing the information on $T$. It differs from the vanilla SCMD in two aspects. Firstly, it computes some average $\bar{\mathbf{w}}_T$ of previous iterates at the $(2^k-1)$-th iteration, $k \in \mathbb{N}$. Secondly, it searches an index $t$ such that $D_\Psi(\bar{\mathbf{w}}_T, \mathbf{w}_t) - D_\Psi(\bar{\mathbf{w}}_T, \mathbf{w}_{t+1}) \leq 2^{1-k}D_\Psi(\bar{\mathbf{w}}_T, \mathbf{w}_T)$ and outputs this individual iterate. Algorithm 2 with $2^k \leq t < 2^{k+1}$ recovers Algorithm 1 with $T = 2^k$ or $T = 2^{k-1}$, depending on whether an index $t \geq 2^k$ satisfying $A_t \leq 2^{1-k}D_\Psi(\bar{\mathbf{w}}_T, \mathbf{w}_T)$ is found or not. Therefore, Algorithm 2 can achieve the same convergence rates as Algorithm 1 in both convex and strongly convex cases. It does not need the information on $T$ and therefore applies to the online learning setting. We refer to Algorithm 2 as Online Composite Mirror Descent with Individual iterates (OCMDI).

## 2.3 Motivation

Before giving theoretical results, we sketch the key **idea** underlying the design of Algorithm 1 which also forms a key foundation of our theoretical analysis. Let $\mathbf{w}^* = \arg\min_{\mathbf{w} \in \mathbb{R}^d} \phi(\mathbf{w})$. Typically we can get the following one-step error bounds for $\{\mathbf{w}_t\}_{t \in \mathbb{N}}$ produced by (2.2) (Lemma A.1 in Appendix)

$$\eta_t \mathbb{E}[\phi(\mathbf{w}_t) - \phi(\mathbf{w})] \leq \mathbb{E}[D_\Psi(\mathbf{w}, \mathbf{w}_t) - D_\Psi(\mathbf{w}, \mathbf{w}_{t+1})] + \eta_t^2 \widetilde{C} \tag{2.3}$$

for any $\mathbf{w} \in \mathbb{R}^d$ independent of $z_t$. Here $\widetilde{C}$ is a constant independent of $t$. If we choose $\mathbf{w} = \mathbf{w}^*$ and can show $\mathbb{E}[D_\Psi(\mathbf{w}^*, \mathbf{w}_t) - D_\Psi(\mathbf{w}^*, \mathbf{w}_{t+1})] = O(\eta_t^2)$, then we would immediately obtain $\mathbb{E}[\phi(\mathbf{w}_t)] - \phi(\mathbf{w}^*) = O(\eta_t)$. This would immediately imply the rate $O(1/\sqrt{t})$ in the convex case and the rate $O(1/t)$ in the strongly convex case since in these two cases the typical step size choices are $\eta_t = 1/\sqrt{t}$ and $\eta_t = 1/t$ (ignoring constant factors), respectively [8]. Since $\mathbf{w}^*$ is unknown, any algorithm requiring an access to $\mathbf{w}^*$ is not implementable. A good surrogate $\bar{\mathbf{w}}_T$ of $\mathbf{w}^*$ should satisfy $\mathbb{E}[\phi(\bar{\mathbf{w}}_T)] - \phi(\mathbf{w}^*) = O(\eta_T)$ to enjoy a tight rate, for which a natural choice should be some average of $\{\mathbf{w}_t\}_{t \leq T}$. Then, we take $\mathbf{w} = \bar{\mathbf{w}}_T$ in (2.3) and need to find an index $T^* \in \{T, T+1, \ldots, 2T-1\}$ such that $\mathbb{E}[D_\Psi(\bar{\mathbf{w}}_T, \mathbf{w}_{T^*}) - D_\Psi(\bar{\mathbf{w}}_T, \mathbf{w}_{T^*+1})] = O(\eta_T^2)$. By the non-negativity of Bregman distance, there always exists a $T^* \in \{T, T+1, \ldots, 2T-1\}$ satisfying (see Lemma A.2 in Appendix)

$$D_\Psi(\bar{\mathbf{w}}_T, \mathbf{w}_{T^*}) - D_\Psi(\bar{\mathbf{w}}_T, \mathbf{w}_{T^*+1}) \leq T^{-1}D_\Psi(\bar{\mathbf{w}}_T, \mathbf{w}_T).$$

This motivates us to search the time index $T^*$ by Algorithm 1 (line 17). It is clear from (2.3) that

$$\mathbb{E}[\phi(\mathbf{w}_{T^*}) - \phi(\bar{\mathbf{w}}_T)] \leq (T\eta_{T^*})^{-1}\mathbb{E}[D_\Psi(\bar{\mathbf{w}}_T, \mathbf{w}_T)] + \eta_{T^*}\widetilde{C}. \tag{2.4}$$

The term $\mathbb{E}[\phi(\mathbf{w}_{T^*})] - \phi(\mathbf{w}^*)$ then can be estimated by the following error decomposition

$$\mathbb{E}[\phi(\mathbf{w}_{T^*})] - \phi(\mathbf{w}^*) = \mathbb{E}[\phi(\mathbf{w}_{T^*}) - \phi(\bar{\mathbf{w}}_T)] + \mathbb{E}[\phi(\bar{\mathbf{w}}_T) - \phi(\mathbf{w}^*)] \tag{2.5}$$

$$\leq (T\eta_{T^*})^{-1}\mathbb{E}[D_\Psi(\bar{\mathbf{w}}_T, \mathbf{w}_T)] + \eta_{T^*}\widetilde{C} + \mathbb{E}[\phi(\bar{\mathbf{w}}_T) - \phi(\mathbf{w}^*)].$$

To derive the desired bound $\mathbb{E}[\phi(\mathbf{w}_{T^*})] - \phi(\mathbf{w}^*) = O(\eta_T)$, it suffices to show

$$(T\eta_{T^*})^{-1}\mathbb{E}[D_\Psi(\bar{\mathbf{w}}_T, \mathbf{w}_T)] = O(\eta_T) \quad \text{and} \quad \mathbb{E}[\phi(\bar{\mathbf{w}}_T) - \phi(\mathbf{w}^*)] = O(\eta_T).$$

More specifically, we need to show $\mathbb{E}[D_\Psi(\bar{\mathbf{w}}_T, \mathbf{w}_T)] = O(1), \mathbb{E}[\phi(\bar{\mathbf{w}}_T) - \phi(\mathbf{w}^*)] = O(T^{-\frac{1}{2}})$ in the convex case with $\eta_t = 1/\sqrt{t}$, and $\mathbb{E}[D_\Psi(\bar{\mathbf{w}}_T, \mathbf{w}_T)] = O(T^{-1}), \mathbb{E}[\phi(\bar{\mathbf{w}}_T) - \phi(\mathbf{w}^*)] = O(T^{-1})$ in the strongly convex case with $\eta_t = 1/t$. We will show this is possible by choosing $T^*$ in Algorithm 1.

## 3 Optimal Convergence Rates

We present here optimal convergence rates for SCMDI for both convex and strongly convex objectives. To this aim, we need to impose some standard assumptions. We assume that the mirror map $\Psi$ is $\sigma_\Psi$-strongly convex w.r.t. a norm $\|\cdot\|$ in the sense $D_\Psi(\mathbf{w}, \tilde{\mathbf{w}}) \geq 2^{-1}\sigma_\Psi \|\mathbf{w} - \tilde{\mathbf{w}}\|^2$ for all $\mathbf{w}, \tilde{\mathbf{w}} \in \mathbb{R}^d$ ($\sigma_\Psi > 0$). We always assume the existence of $A$ and $B > 0$ such that ($\|\cdot\|_*$ is the dual norm of $\|\cdot\|$)

$$\|f'(\mathbf{w}, z)\|_*^2 \leq Af(\mathbf{w}, z) + B \text{ and } \|r'(\mathbf{w})\|_*^2 \leq Ar(\mathbf{w}) + B \tag{3.1}$$

for any $\mathbf{w} \in \mathbb{R}^d, z \in \mathcal{Z}$ and any $f'(\mathbf{w}, z) \in \partial f(\mathbf{w}, z), r'(\mathbf{w}) \in \partial r(\mathbf{w})$. Many popular learning methods use loss functions and regularizers satisfying (3.1) [7, 44]. For example, if $|\ell'(a, y)|^2 \leq \tilde{A}\ell(a, y) + \tilde{B}$ for some $\tilde{A}, \tilde{B} > 0$ and all $a, y$, then $f(\mathbf{w}, z) = \ell(\langle \mathbf{w}, x \rangle, y)$ would satisfy the first inequality of (3.1) if $\mathcal{X}$ is bounded. Here $\ell'(a, y)$ denotes a subgradient of $\ell$ w.r.t. the first argument. Examples of such $\ell$ include all smooth functions and all Lipschitz continuous functions widely used in machine learning [42, 44]. Examples of $r$ satisfying (3.1) include $r(\mathbf{w}) = \lambda\|\mathbf{w}\|_p^p$ with $p \in [1, 2]$. We say $\Psi$ is $L_\Psi$-smooth w.r.t. $\|\cdot\|$ if $D_\Psi(\mathbf{w}, \tilde{\mathbf{w}}) \leq \frac{L_\Psi}{2}\|\mathbf{w} - \tilde{\mathbf{w}}\|^2$ for all $\mathbf{w}, \tilde{\mathbf{w}} \in \mathbb{R}^d$.

We always assume the existence of $\sigma_F, \sigma_r \geq 0$ such that for all $\mathbf{w}, \tilde{\mathbf{w}} \in \mathbb{R}^d$

$$F(\mathbf{w}) - F(\tilde{\mathbf{w}}) - \langle \mathbf{w} - \tilde{\mathbf{w}}, F'(\tilde{\mathbf{w}})\rangle \geq \sigma_F D_\Psi(\mathbf{w}, \tilde{\mathbf{w}}), \ r(\mathbf{w}) - r(\tilde{\mathbf{w}}) - \langle \mathbf{w} - \tilde{\mathbf{w}}, r'(\tilde{\mathbf{w}})\rangle \geq \sigma_r D_\Psi(\mathbf{w}, \tilde{\mathbf{w}}) \tag{3.2}$$

For $\sigma_\phi := \sigma_F + \sigma_r$, the cases $\sigma_\phi = 0$ and $\sigma_\phi > 0$ correspond to convex and strongly convex objectives, respectively.

### 3.1 Convex objectives

Our first result is an optimal rate $O(T^{-\frac{1}{2}})$ for convex objectives under a boundedness assumption of iterates. The boundedness assumptions $\mathbb{E}[D_\Psi(\mathbf{w}^*, \mathbf{w}_t)] \leq D, \forall t \in \mathbb{N}$, imposed also in the literature [7, 8, 37], always hold for the regularizer of the form $r(\mathbf{w}) = I_{\mathcal{W}_0}(\mathbf{w}) + \tilde{r}(\mathbf{w})$, where $\mathcal{W}_0$ is a convex and compact domain, $I_{\mathcal{W}_0}$ is an indicator function with $I_{\mathcal{W}_0}(\mathbf{w}) = 0$ if $\mathbf{w} \in \mathcal{W}_0$ and $\infty$ otherwise, and $\tilde{r} : \mathbb{R}^d \to \mathbb{R}_+$ is convex. Proofs of theoretical results in this subsection are given in Appendix B.

**Theorem 1.** *Let $D > 0$. Assume $\mathbb{E}[D_\Psi(\mathbf{w}^*, \mathbf{w}_t)] \leq D$ for all $t \in \mathbb{N}$ and $\mathbb{E}[D_\Psi(\bar{\mathbf{w}}_T, \mathbf{w}_T)] \leq D$. Let $\mathbf{w}_{T^*}$ be defined by Algorithm 1 with $\eta_t = \mu/\sqrt{t}$ satisfying $\mu \leq \sigma_\Psi(2A)^{-1}$. Then, $\mathbb{E}[\phi(\mathbf{w}_{T^*}) - \phi(\mathbf{w}^*)] \leq \frac{\tilde{C}_1}{\sqrt{T}}$, where $\tilde{C}_1 = (4 + 2\sqrt{2})\mu^{-1}D + 10\mu\sigma_\Psi^{-1}(A\phi(\mathbf{w}^*) + 2B)$.*

Theorem 1 requires to impose a boundedness assumption on iterates, which is removed in the following theorem on convergence rates. The assumption $D_\Psi(\mathbf{w}, \tilde{\mathbf{w}}) \leq L_\Psi\|\mathbf{w} - \tilde{\mathbf{w}}\|^\alpha$ is milder than assuming a smoothness of $\Psi$, the latter of which is satisfied if $\Psi = \Psi_2$.

**Theorem 2.** *Let $\mathbf{w}_{T^*}$ be produced by Algorithm 1 with a non-increasing step size sequence and $\eta_1 \leq \sigma_\Psi(2A)^{-1}$. Assume that there exists $\alpha \in [0, 2]$ and $L_\Psi > 0$ such that $D_\Psi(\mathbf{w}, \tilde{\mathbf{w}}) \leq L_\Psi\|\mathbf{w} - \tilde{\mathbf{w}}\|^\alpha$ for all $\mathbf{w}, \tilde{\mathbf{w}} \in \mathbb{R}^d$. Let $D_T = D_\Psi(\mathbf{w}^*, \mathbf{w}_1) + \sigma_\Psi^{-1}(A\phi(\mathbf{w}^*) + 2B)\sum_{t=1}^T \eta_t^2$. Then,*

$$\mathbb{E}[\phi(\mathbf{w}_{T^*})] - \phi(\mathbf{w}^*) \leq \frac{2L_\Psi[4\sigma_\Psi^{-1}\alpha D_T + 1] + 4D_T}{T\eta_{2T}} + \frac{2A\phi(\mathbf{w}^*) + 4B}{\sigma_\Psi}\Big[\eta_T + \frac{2}{T}\sum_{t=1}^T \eta_t\Big]. \tag{3.3}$$

**Corollary 3.** *Suppose the assumptions in Theorem 2 hold. (a) If $\lim_{t\to\infty} \frac{\sum_{\tilde{t}=1}^t \eta_{\tilde{t}}^2}{t\eta_t} = 0$, then $\lim_{T\to\infty} \mathbb{E}[\phi(\mathbf{w}_{T^*})] = \phi(\mathbf{w}^*)$. (b) If $\eta_t = \frac{\mu}{\sqrt{T}}$ for $t = 1, \ldots, 2T$, then $\mathbb{E}[\phi(\mathbf{w}_{T^*})] - \phi(\mathbf{w}^*) = O(T^{-\frac{1}{2}})$.*

### 3.2 Strongly convex objectives

We now turn to Algorithm 1 for strongly convex objectives, towards which a first step is the following theorem on the performance of vanilla SCMD. Theorem 4 shows that both the suboptimality of some weighted averaged iterates and the Bregman distance between the last iterate and $\mathbf{w}^*$ converge with the rate $O(T^{-1})$. Theorem 4 is an extension of the results in [20] on the projected SGD to SCMD. Discussions in [20] require to impose a boundedness assumption on subgradients as $\mathbb{E}_Z[\|f'(\mathbf{w}_t, Z)\|_2^2] \leq \tilde{D}$ for some $\tilde{D} > 0$ and all $t \in \mathbb{N}$. Indeed, an independent section was included in [20] to show that this boundedness assumption holds for an SVM-like objective. Theorem 4 shows that we can derive the same convergence rate without this boundedness assumption and

is therefore more applicable. Theorem 5 gives a sufficient condition on step size sequences for the convergence. Upper and lower bounds on convergence rates are established in Theorem 6 and Theorem 7, respectively. The proofs of theoretical results in this subsection are given in Appendix C. If $\sigma_r = 0$, then $\bar{\mathbf{w}}_T$ defined in Theorem 4 can be simplified as $\bar{\mathbf{w}}_T = \left[\sum_{t=1}^{T}(t+1)\right]^{-1}\sum_{t=1}^{T}(t+1)\mathbf{w}_t$.

**Theorem 4.** *Assume (3.2) holds with $\sigma_\phi > 0$. Let $\{\mathbf{w}_t\}_{t\in\mathbb{N}}$ be generated by (2.2) with $\eta_t = \frac{2}{\sigma_\phi t + 2\sigma_F}$. Define*

$$\bar{\mathbf{w}}_T = \Big[\sum_{t=1}^{T}(t+1)(t+2)\eta_t\Big]^{-1}\sum_{t=1}^{T}(t+1)(t+2)\eta_t\mathbf{w}_t.$$

*Then, there exists a constant $\widetilde{C}_2 > 0$ independent of $T$ and $\sigma_\phi$ (explicitly given in the proof) such that*

$$\mathbb{E}[\phi(\bar{\mathbf{w}}_T) - \phi(\mathbf{w}^*)] \leq \frac{4\widetilde{C}_2}{\sigma_\phi(T+1)} \quad and \quad \mathbb{E}[D_\Psi(\mathbf{w}^*, \mathbf{w}_{T+1})] \leq \frac{\widetilde{C}_2}{\sigma_\phi^2(T+2)}. \qquad (3.4)$$

**Theorem 5.** *Assume (3.2) holds with $\sigma_\phi > 0$. Let $\{\mathbf{w}_t\}_{t\in\mathbb{N}}$ be generated by (2.2). If $\lim_{t\to\infty}\eta_t = 0$ and $\sum_{t=1}^{\infty}\eta_t = \infty$, then $\lim_{T\to\infty}\mathbb{E}[D_\Psi(\mathbf{w}^*, \mathbf{w}_T)] = 0$.*

With Theorem 4 at hand, we can provide optimal rates for the output produced by Algorithm 1.

**Theorem 6.** *Assume (3.2) holds with $\sigma_\phi > 0$. Let $\mathbf{w}_{T^*}$ be generated by Algorithm 1 with $\eta_t = \frac{2}{\sigma_\phi t + 2\sigma_F}$. Assume that $\Psi$ is $L_\Psi$-strongly smooth ($L_\Psi > 0$). If $T \geq \frac{4A}{\sigma_\Psi\sigma_\phi}$, then there exists a constant $\widetilde{C}_3 > 0$ independent of $T$ and $\sigma_\phi$ (explicitly given in the proof) such that $\mathbb{E}\big[\phi(\mathbf{w}_{T^*}) - \phi(\mathbf{w}^*)\big] \leq \frac{\widetilde{C}_3}{T\sigma_\phi}$.*

Theorem 7 presents lower bounds matching the above upper bounds up to a constant factor, which shows that the selected iterate has achieved the best possible rate and there are no stronger results. The matching upper and lower bounds here apply to the specific SGD and specific optimization problems, while minimax bounds is related to the existence of a problem for any algorithm [1]. Similar results were derived in [21] for online mirror descent. Here we give a different and simpler proof.

**Theorem 7.** *Let $\{\mathbf{w}_t\}_t$ be the sequence produced by (2.2) with $\Psi(\mathbf{w}) = \frac{1}{2}\|\mathbf{w}\|_2^2$ and $r(\mathbf{w}) = 0$. Suppose $f$ is differentiable, $\phi$ is $L_\phi$-smooth w.r.t. $\|\cdot\|_2$ and $\eta_t \leq 1/(2L_\phi)$. If for any $\mathbf{w} \in \mathbb{R}^d$, $\mathbb{E}[\|\nabla f(\mathbf{w}, z) - \nabla F(\mathbf{w})\|_2^2] \geq \sigma^2$ for some $\sigma > 0$, then*

$$\mathbb{E}[\|\mathbf{w}_{t+1} - \mathbf{w}^*\|_2^2] \geq \min\{\|\mathbf{w}_1 - \mathbf{w}^*\|_2^2, \eta_1\sigma^2/(2L_\phi), \ldots, \eta_t\sigma^2/(2L_\phi)\}.$$

## 4 Discussions

We discuss related work on stochastic/online learning algorithms with different averaging schemes.

A very common scheme is to output an average of all iterates with uniform weights (UNI-AVE) [44, 46], which is able to attain the optimal rate $O(1/\sqrt{T})$ and the suboptimal rate $O(T^{-1}\log T)$ in the convex and strongly convex case, respectively [8]. A counterpart is established to show that the rate $O(T^{-1}\log T)$ is the best possible one for this simple averaging scheme [31].

For strongly convex objectives, the first algorithm able to attain the optimal rate $O(T^{-1})$ is the epoch-GD algorithm proposed in [14], which performs stochastic mirror descent in each epoch with a fixed step size. The step size decreases exponentially in each consecutive epoch and the averaged iterate of the last epoch is outputted as the solution. Since then, several other averaging schemes were developed to attain optimal rates, including a suffix-averaging scheme (SUFFIX) returning the uniform average of the last half of SGD iterates [31], a weighted averaging scheme (WEI-AVE) with a weight of $t + 1$ for $\mathbf{w}_t$ [20] and a polynomial-decay averaging scheme [37].

Motivated by the side effects of averaging in either destroying the sparsity of the solution or slowing down the training speed [31], the property of the last iterate (LAST) has also received a lot of attention. For smooth objective functions, SGD with the last iterate can achieve the optimal rate [31]. The analysis of last iterate for non-smooth objective functions is much more challenging, for which an interesting technique to relate the last iterate with suffix-average of iterates was developed [37, 44]. Based on this, rates $O(T^{-\frac{1}{2}}\log T)$ and $O(T^{-1}\log T)$ were established in the convex and strongly-convex cases [23, 37], respectively. These results motivate a natural question of whether the last iterate achieves the optimal rate in the non-smooth scenario. This open question was resolved in [13]

by constructing a problem for which SGD with the last iterate achieves at most the rate $\Omega(T^{-1}\log T)$. An interesting probabilistic rate $O(T^{-1}\log T)$ was also developed for SGD with the last iterate [13].

All the above mentioned methods require an averaging scheme to attain the optimal convergence rates, which, however, may destroy the sparsity of solutions and slow down the practical training speed. A simple scheme to preserve the sparsity of solutions while still enjoying the optimal rate is to draw an individual iterate *randomly* from the iterate sequence by (2.2) [10, 16]. Indeed, one can randomly draw the output according to a distribution over the iterate sequence with the probability mass function determined by any optimal weighted averaging scheme [20, 31, 37]. However, this scheme introduces new variances due to the choice of a random iterate as the output. Moreover, as we will verify in experiments, this scheme can slow down the practical training speed since the randomly selected iterate is not necessarily to be a favorable one in the iterate sequence. Furthermore, it requires to either store all the iterate sequence or determine the number of iterations beforehand, which does not apply to the online learning scenario. After the acceptance of the paper, we noticed an interesting step-size modification scheme to get optimal convergence rates [15]. However, the step-size scheme there requires to use the information of $T$, and therefore is not applicable to the online learning scenario. Furthermore, the algorithm and optimal convergence rates there are developed for the particular $T$-th iterate and not for other iterates.

In this paper, we propose a novel stochastic/online learning algorithm able to achieve optimal rates. As compared to the averaging scheme in [14, 20, 31, 37], our scheme of outputting an individual iterate is able to preserve the sparsity structure of solutions. As compared to the scheme of randomly choosing an individual iterate, our method avoids the added variance as well as the drawback of slowing down the training speed due to the random iterate index. Our idea is not to output the last iterate but to select an individual iterate based on a careful analysis on the one-step progress bound sketched in Section 2.3. Indeed, by (2.3), the quality of $\mathbf{w}_t$ depends on $A_t = D_\Psi(\bar{\mathbf{w}}_T, \mathbf{w}_t) - D_\Psi(\bar{\mathbf{w}}_T, \mathbf{w}_{t+1})$. The smaller the $A_t$, the better the quality. This motivates us to select a $T^*$ with small $A_{T^*}$. Intuitively, $A_t$ is related to the distance between $\mathbf{w}_t$ and $\mathbf{w}_{t+1}$. Therefore, our selection of the iterate shares some spirit with the widely used heuristic of terminating the algorithm when the successive iterates are close. Our analysis also refines existing studies of optimal rates of SGD by removing the boundedness assumption on subgradients [20, 31, 37]. It should be mentioned that the boundedness assumption was relaxed as $\mathbb{E}_z[\|f'(\mathbf{w}_t, z)\|_*^2] \leq \widetilde{A} + \widetilde{B}\|F'(\mathbf{w}_t)\|_*^2$ in [4] for $\widetilde{A}, \widetilde{B} > 0$ and removed in [27], the latter of which, however, requires the objective function to be smooth.

## 5  Experimental Results

In this section, we justify the effectiveness of our algorithm by presenting experimental comparisons with the following averaging strategies: WEI-AVE [20], UNI-AVE, LAST, SUFFIX [31] and RAND (outputting a random iterate chosen from the uniform distribution over the last half of iterates). We consider two applications: binary classification and tomography reconstruction in image processing.

### 5.1  Binary classification

We first consider SVM models with linear kernels. We use 16 real-world datasets whose information is summarized in Table C.1 in Appendix D.1[3]. For each dataset, we use 80 percents of the data for training and reserve the remaining 20 percents for testing. The objective function we consider for a training dataset $\{(x_1, y_1), \ldots, (x_n, y_n)\}$ is

$$\phi(\mathbf{w}) = \frac{\lambda}{2}\|\mathbf{w}\|_2^2 + \frac{1}{n}\sum_{i=1}^{n}\max\{0, 1 - y_i\langle\mathbf{w}, x_i\rangle\},$$

which is $\lambda$-strongly convex w.r.t. $\|\cdot\|_2$. Analogous to [20], we set $\lambda$ to the reciprocal of training sample size. For datasets with multiple class labels, we group the first half of labels into the positive label, and the second half into the negative label. We consider step sizes of the form $\eta_t = \mu/(\lambda t)$ and tune the parameter $\mu$ in the set $2^{\{-12,-11,\ldots,4\}}$ by 10-fold cross validation. We repeat the experiment 40 times and report the average of experimental results. We consider two approaches to optimize the objective function $\phi$ by whether to separate the regularizer and the data-fitting term. In the first approach, we apply (2.2) with $\Psi(\mathbf{w}) = \frac{1}{2}\|\mathbf{w}\|_2^2, f(\mathbf{w}, z) = \frac{\lambda}{2}\|\mathbf{w}\|_2^2 + \max\{0, 1 - y\langle\mathbf{w}, x\rangle\}$ and $r(\mathbf{w}) = 0$,

which is just the SGD. In the second approach, we apply (2.2) with $\Psi(\mathbf{w}) = \frac{1}{2}\|\mathbf{w}\|_2^2, f(\mathbf{w}, z) = \max\{0, 1 - y\langle\mathbf{w}, x\rangle\}$ and $r(\mathbf{w}) = \frac{\lambda}{2}\|\mathbf{w}\|_2^2$, which is a stochastic proximal gradient descent (SPGD). In Figure 1 and Figure 2, we plot the objective function values on the testing dataset against iteration numbers for SGD and SPGD, respectively. It is clear that UNI-AVE is always the worst strategy in our experiments. WEI-AVE assigns more weights to recent iterates and can improve the performance, which is further outperformed by SUFFIX. RAND fluctuates a bit since the randomly selected index at the $t$-th iteration may not be an increasing function of $t$. LAST is overwritten by OCMDI due to their similar behavior, which behave best in our experiments especially in the beginning of the optimization process. The similarity between OCMDI and LAST can be explained as follows. If $A_t$ (defined in Section 4) is small then it is likely that $A_{t+1}$ is also small. Therefore, our algorithm is prone to select the last part of iterates. We also plot the objective function values on training datasets against $T$, which show similar behavior and are deferred to Figures D.4, D.3 in Appendix.

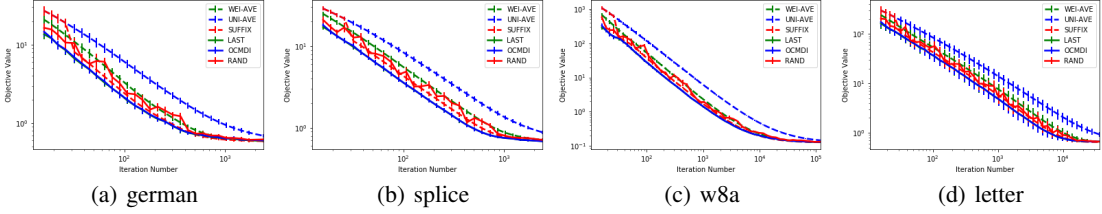

(a) german        (b) splice        (c) w8a        (d) letter

Figure 1: Objective function values on test datasets versus iteration numbers for SGD.

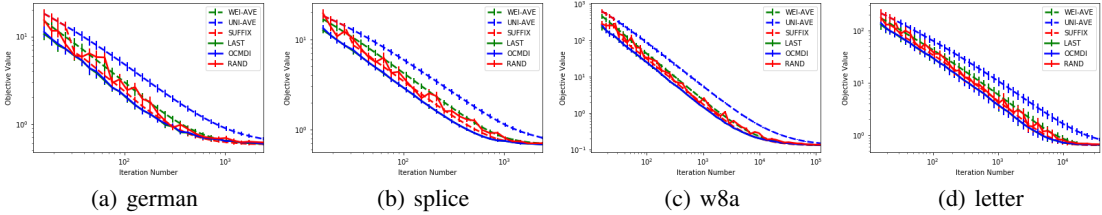

(a) german        (b) splice        (c) w8a        (d) letter

Figure 2: Objective function values on test datasets versus iteration numbers for SPGD.

## 5.2 Tomography reconstruction

We now consider tomography reconstruction in image processing. We use AIR toolbox [12] to create a CT-measurement matrix $A \in \mathbb{R}^{n \times d}$, an output vector $\mathbf{y}^\dagger \in \mathbb{R}^n$ and a $N \times N$ sparse image encoded by a vector $\mathbf{w}^\dagger \in \mathbb{R}^d$ with $d = N^2$. Each row of $A$ together with the corresponding row in $\mathbf{y}$ indicates a line integral from a fan bean projection geometry. Therefore, the true image $\mathbf{w}^\dagger$ satisfies $A\mathbf{w}^\dagger = \mathbf{y}^\dagger$. We consider a noisy case by adding Gaussian noise $N(0, (0.05|y_i^\dagger|)^2)$ to $y_i^\dagger$ and get the noisy output $\mathbf{y}$. Our aim is to reconstruct the image $\mathbf{w}^\dagger$ from the matrix $A$ and the noisy measurements $\mathbf{y}$ by finding an approximate solution of the equation $A\mathbf{w} = \mathbf{y}$. Since many components of the true image $\mathbf{w}^\dagger$ varnish, we apply randomized sparse Kaczmarz algorithm, which is a simple and efficient algorithm to generate sparse approximate solutions to linear systems [5, 34]. Let $\epsilon > 0, \lambda > 0$ be two parameters and consider the mirror map $\Psi^{(\epsilon,\lambda)}(\mathbf{w}) = \lambda \sum_{i=1}^d g_\epsilon(w(i)) + \frac{1}{2}\|\mathbf{w}\|_2^2$, where $g_\epsilon(\xi) = \frac{\xi^2}{2\epsilon}$ for $|\xi| \leq \epsilon$ and $|\xi| - \frac{\epsilon}{2}$ for $|\xi| > \epsilon$. At each iteration $t$, we randomly select an index $i_t$ from the uniform distribution over $\{1, \dots, n\}$ and denote $x_{i_t} = A_{i_t}^\top$, where $A_{i_t}^\top$ is the transpose of $A_{i_t}$. Given $\mathbf{w}_1 \in \mathbb{R}^d$ and $\mathbf{v}_1 = \nabla\Psi^{(\epsilon,\lambda)}(\mathbf{w}_1)$, the randomized sparse Kaczmarz algorithm updates the model as

$$\mathbf{v}_{t+1} = \mathbf{v}_t - \eta_t(\langle\mathbf{w}_t, x_{i_t}\rangle - y_{i_t})x_{i_t}, \quad \mathbf{w}_{t+1} = S_{\lambda,\epsilon}(\mathbf{v}_{t+1}), \tag{5.1}$$

where $S_{\lambda,\epsilon} : \mathbb{R}^d \mapsto \mathbb{R}^d$ is defined component-wisely by the soft-thresholding function $S_{\lambda,\epsilon} : \mathbb{R} \mapsto \mathbb{R}$ given as [5]

$$S_{\lambda,\epsilon}(\xi) := \begin{cases} (\xi\epsilon)/(\lambda + \epsilon), & \text{if } |\xi| \leq \lambda + \epsilon \\ \text{sgn}(\xi)\max(|\xi| - \lambda, 0), & \text{otherwise.} \end{cases}$$

Here $\text{sgn}(a)$ denotes the sign of $a \in \mathbb{R}$. Algorithm (5.1) can be equivalently formulated as [22]

$$\mathbf{w}_{t+1} = \arg\min_{\mathbf{w} \in \mathbb{R}^d} \eta_t\langle\mathbf{w} - \mathbf{w}_t, (\langle\mathbf{w}_t, x_{i_t}\rangle - y_{i_t})x_{i_t}\rangle + D_{\Psi^{(\epsilon,\lambda)}}(\mathbf{w}, \mathbf{w}_t).$$

It is clear from the above equivalent formulation that (5.1) is an instantiation of (2.2) with $f(\mathbf{w}, z) = \frac{1}{2}(\langle \mathbf{w}, x \rangle - y)^2$, $r(\mathbf{w}) = 0$ and $\Psi(\mathbf{w}) = \Psi^{(\epsilon, \lambda)}(\mathbf{w})$ to minimize $F(\mathbf{w}) = \frac{1}{n}\|A\mathbf{w} - \mathbf{y}\|_2^2$. It was shown that $F(\mathbf{w})$ satisfies (3.2) with $\sigma_F = \sigma_{\min}(A^\top A / n)$, $\mathbf{w} = \mathbf{w}^*$ and $\tilde{\mathbf{w}} = \mathbf{w}_t, t \in \mathbb{N}$ [22], where $\sigma_{\min}(\widetilde{A})$ denotes the minimal positive eigenvalue of a matrix $\widetilde{A}$. Therefore, our theoretical analysis in Section 3.2 applies[4]. We randomly choose $\mathbf{w}_1$ from the uniform distribution in $[0, 1]^d$ and set $\lambda = 1, \epsilon = 10^{-8}$ as suggested in [5]. We repeat the experiment 40 times and report the average of results.

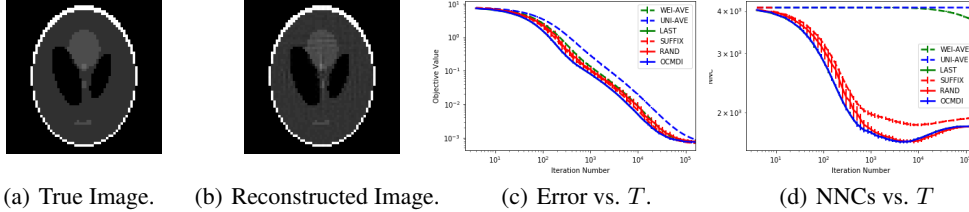

(a) True Image.     (b) Reconstructed Image.     (c) Error vs. $T$.     (d) NNCs vs. $T$

Figure 3: Tomography reconstruction with $N = 64, n = 23040$ and $5\%$ relative noise. Panel (a) and (b) are the true image and the reconstructed image by OCMDI, respectively. Panel (c) and (d) plot the errors and NNCs versus iteration numbers.

In Figure 3, we present results with $N = 64, d = 4096$ and $n = 23040$. Panels (a) and (b) display the true image and the image output by OCMDI, respectively. In Panels (c) and (d), we plot the errors and the number of non-zero components (NNCs) for the output of algorithms with different averaging strategies. According to Figure 3, LAST is overwritten by OCMDI since they behave analogously, both of which achieve the fastest training speed among all considered methods. RAND is outperformed by SUFFIX in terms of training speed. Moreover, OCMDI, RAND and LAST can produce much more sparse solutions than other methods. Indeed, the true image $\mathbf{w}^\dagger$ has 1686 non-zero components, while UNI-AVE, WEI-AVE, SUFFIX, RAND and OCMDI produce outputs with 4096, 3727, 1943, 1823 and 1828 non-zero components, respectively. The advantage of OCMDI over LAST is that OCMDI can theoretically achieve optimal convergence rate. Similar phenomenon also appears for the case with $N = 32$ and $n = 11520$, which we plot in Figure D.5 in Appendix.

# 6   Conclusion

We propose a novel variant of SCMD with optimal convergence rates. An advantage of this algorithm over existing optimal learning algorithms is that it outputs an individual iterate without a random selection of index, which is able to preserve the sparsity structure without slowing down the training speed. Experimental results in both the domain of binary classification and tomography reconstruction demonstrate the ability of our algorithm in getting a fast training speed as well as in producing sparse solutions. Some interesting work include a theoretical justification on the advantage of sparsity of our method over other methods [11, 38] and an extension of our analysis to non-convex problems [10, 32, 45].

**Acknowledgments**

The work of Y. Lei, P. Yang and K. Tang is supported partially by the National Key Research and Development Program of China (Grant No. 2017YFB1003102), the National Natural Science Foundation of China (Grant Nos. 61806091, 61806090 and 61672478), the Program for University Key Laboratory of Guangdong Province (Grant No. 2017KSYS008) and Shenzhen Peacock Plan (Grant No. KQTD2016112514355531). The work of D.-X. Zhou is supported partially by the Research Grants Council of Hong Kong [Project No. CityU 11338616] and by National Nature Science Foundation of China [Grant No. 11671307]. Y. Lei also acknowledges a Humboldt Research Fellowship from the Alexander von Humboldt Foundation.

## Footnotes

[2]When mentioning averaging, individual iterates or last iterate, our attention is with respect to the iterate sequence produced by (2.2) below.

[3]We display experimental results for 4 datasets here due to space limit. Complete results are in Appendix.

[4]In our analysis, we only use (3.2) for $\mathbf{w} = \mathbf{w}^*$ and $\tilde{\mathbf{w}} = \mathbf{w}_t$ (a restricted strong convexity in literature).

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
