[Supplementary Material · appendix.pdf]

# Supplementary Material for "Optimal Stochastic and Online Learning with Individual Iterates"

**Yunwen Lei**[1,2]  **Peng Yang**[1]  **Ke Tang**[1]*  **Ding-Xuan Zhou**[3]

[1]University Key Laboratory of Evolving Intelligent Systems of Guangdong Province,
Department of Computer Science and Engineering,
Southern University of Science and Technology, Shenzhen 518055, China

[2]Department of Computer Science,
Technical University of Kaiserslautern, Kaiserslautern 67653, Germany

[3]School of Data Science and Department of Mathematics,
City University of Hong Kong, Kowloon, Hong Kong, China

{leiyw, yangp, tangk3}@sustech.edu.cn   mazhou@cityu.edu.hk

## A  Motivation of Algorithms

In this section, we provide rigorous theoretical results on the motivation of algorithms presented in Section 2.3. The first result (Lemma A.1) is a standard one-step progress bound which quantifies how the iterate would move towards $\mathbf{w}^*$ with a single update. The second result (Lemma A.2) shows that the time index $T^*$ in Algorithm 1 can be always found. Therefore, the algorithm is well defined.

**Lemma A.1.** *Let the sequence $\{\mathbf{w}_t\}_{t\in\mathbb{N}}$ be generated by (2.2), then the following inequality holds for any $\mathbf{w} \in \mathbb{R}^d$ independent of $z_t$*

$$\mathbb{E}_{z_t}[D_\Psi(\mathbf{w}, \mathbf{w}_{t+1})] - D_\Psi(\mathbf{w}, \mathbf{w}_t) \leq \eta_t\big[\phi(\mathbf{w}) - \phi(\mathbf{w}_t)\big] + \sigma_\Psi^{-1}\eta_t^2\big[A\phi(\mathbf{w}_t) + 2B\big]$$
$$- \eta_t\Big[\sigma_F D_\Psi(\mathbf{w}, \mathbf{w}_t) + \sigma_r \mathbb{E}_{z_t}[D_\Psi(\mathbf{w}, \mathbf{w}_{t+1})]\Big]. \quad \text{(A.1)}$$

*Proof.* According to the first-order optimality condition in (2.2), there exists an $r'(\mathbf{w}_{t+1}) \in \partial r(\mathbf{w}_{t+1})$ satisfying

$$\eta_t f'(\mathbf{w}_t, z_t) + \eta_t r'(\mathbf{w}_{t+1}) + \nabla\Psi(\mathbf{w}_{t+1}) - \nabla\Psi(\mathbf{w}_t) = 0,$$

from which and the identity [1]

$$D_\Psi(\mathbf{w}, \mathbf{w}_{t+1}) + D_\Psi(\mathbf{w}_{t+1}, \mathbf{w}_t) - D_\Psi(\mathbf{w}, \mathbf{w}_t) = \langle \mathbf{w} - \mathbf{w}_{t+1}, \nabla\Psi(\mathbf{w}_t) - \nabla\Psi(\mathbf{w}_{t+1})\rangle$$

we derive

$$D_\Psi(\mathbf{w}, \mathbf{w}_{t+1}) - D_\Psi(\mathbf{w}, \mathbf{w}_t) = D_\Psi(\mathbf{w}, \mathbf{w}_{t+1}) + D_\Psi(\mathbf{w}_{t+1}, \mathbf{w}_t) - D_\Psi(\mathbf{w}, \mathbf{w}_t) - D_\Psi(\mathbf{w}_{t+1}, \mathbf{w}_t)$$
$$= \langle \mathbf{w} - \mathbf{w}_{t+1}, \nabla\Psi(\mathbf{w}_t) - \nabla\Psi(\mathbf{w}_{t+1})\rangle - D_\Psi(\mathbf{w}_{t+1}, \mathbf{w}_t)$$
$$= \eta_t\langle \mathbf{w} - \mathbf{w}_{t+1}, f'(\mathbf{w}_t, z_t) + r'(\mathbf{w}_{t+1})\rangle - D_\Psi(\mathbf{w}_{t+1}, \mathbf{w}_t)$$
$$\leq \eta_t\langle \mathbf{w} - \mathbf{w}_{t+1}, f'(\mathbf{w}_t, z_t)\rangle + \eta_t\big[r(\mathbf{w}) - r(\mathbf{w}_{t+1}) - \sigma_r D_\Psi(\mathbf{w}, \mathbf{w}_{t+1})\big] - D_\Psi(\mathbf{w}_{t+1}, \mathbf{w}_t)$$
$$= \eta_t\langle \mathbf{w} - \mathbf{w}_t, f'(\mathbf{w}_t, z_t)\rangle + \eta_t\langle \mathbf{w}_t - \mathbf{w}_{t+1}, f'(\mathbf{w}_t, z_t)\rangle + \eta_t[r(\mathbf{w}) - r(\mathbf{w}_t)]$$
$$+ \eta_t[r(\mathbf{w}_t) - r(\mathbf{w}_{t+1})] - \sigma_r\eta_t D_\Psi(\mathbf{w}, \mathbf{w}_{t+1}) - D_\Psi(\mathbf{w}_{t+1}, \mathbf{w}_t). \quad \text{(A.2)}$$

---

Here, we have used the $\sigma_r$-strong convexity of $r$ (3.2) in the inequality. From the convexity of $r$, the definition of dual norm and the strong convexity of $\Psi$, it follows that

$$
\begin{aligned}
&\eta_t\big[\langle \mathbf{w}_t - \mathbf{w}_{t+1}, f'(\mathbf{w}_t, z_t)\rangle + r(\mathbf{w}_t) - r(\mathbf{w}_{t+1})\big] - D_\Psi(\mathbf{w}_{t+1}, \mathbf{w}_t) \\
&\leq \eta_t \|\mathbf{w}_t - \mathbf{w}_{t+1}\| \|f'(\mathbf{w}_t, z_t)\|_* + \eta_t \langle \mathbf{w}_t - \mathbf{w}_{t+1}, r'(\mathbf{w}_t)\rangle - 2^{-1}\sigma_\Psi \|\mathbf{w}_t - \mathbf{w}_{t+1}\|^2 \\
&\leq \eta_t \|\mathbf{w}_t - \mathbf{w}_{t+1}\| \big[\|f'(\mathbf{w}_t, z_t)\|_* + \|r'(\mathbf{w}_t)\|_*\big] - 2^{-1}\sigma_\Psi \|\mathbf{w}_t - \mathbf{w}_{t+1}\|^2 \\
&\leq 2^{-1}\sigma_\Psi \|\mathbf{w}_t - \mathbf{w}_{t+1}\|^2 + 2^{-1}\sigma_\Psi^{-1}\eta_t^2 \big[\|f'(\mathbf{w}_t, z_t)\|_* + \|r'(\mathbf{w}_t)\|_*\big]^2 - 2^{-1}\sigma_\Psi \|\mathbf{w}_t - \mathbf{w}_{t+1}\|^2 \\
&\leq \sigma_\Psi^{-1}\eta_t^2 \big[\|f'(\mathbf{w}_t, z_t)\|_*^2 + \|r'(\mathbf{w}_t)\|_*^2\big] \leq \sigma_\Psi^{-1}\eta_t^2 \big[Af(\mathbf{w}_t, z_t) + Ar(\mathbf{w}_t) + 2B\big],
\end{aligned}
$$

where we have used the elementary inequality $(a + b)^2 \leq 2(a^2 + b^2)$ and (3.1) in the last two inequalities. Plugging the above inequality back into (A.2), we get

$$
\begin{aligned}
D_\Psi(\mathbf{w}, \mathbf{w}_{t+1}) - D_\Psi(\mathbf{w}, \mathbf{w}_t) &\leq \eta_t \langle \mathbf{w} - \mathbf{w}_t, f'(\mathbf{w}_t, z_t)\rangle + \eta_t[r(\mathbf{w}) - r(\mathbf{w}_t)] \\
&\quad + \sigma_\Psi^{-1}\eta_t^2 \big[Af(\mathbf{w}_t, z_t) + Ar(\mathbf{w}_t) + 2B\big] - \sigma_r\eta_t D_\Psi(\mathbf{w}, \mathbf{w}_{t+1}).
\end{aligned}
$$

Taking conditional expectations with respect to $z_t$ over both sides and using the $\sigma_F$-strong convexity of $F$ (3.2) then give (note both $\mathbf{w}$ and $\mathbf{w}_t$ are independent of $z_t$)

$$
\begin{aligned}
&\mathbb{E}_{z_t}[D_\Psi(\mathbf{w}, \mathbf{w}_{t+1})] - D_\Psi(\mathbf{w}, \mathbf{w}_t) + \sigma_r\eta_t \mathbb{E}_{z_t}[D_\Psi(\mathbf{w}, \mathbf{w}_{t+1})] \\
&\leq \eta_t \langle \mathbf{w} - \mathbf{w}_t, F'(\mathbf{w}_t)\rangle + \eta_t[r(\mathbf{w}) - r(\mathbf{w}_t)] + \sigma_\Psi^{-1}\eta_t^2 \big[AF(\mathbf{w}_t) + Ar(\mathbf{w}_t) + 2B\big] \\
&\leq \eta_t \big[F(\mathbf{w}) - F(\mathbf{w}_t) - \sigma_F D_\Psi(\mathbf{w}, \mathbf{w}_t)\big] + \eta_t[r(\mathbf{w}) - r(\mathbf{w}_t)] + \sigma_\Psi^{-1}\eta_t^2 \big[A\phi(\mathbf{w}_t) + 2B\big].
\end{aligned}
$$

This gives the stated inequality and completes the proof. $\qquad\square$

**Lemma A.2.** *There exists an $t \in \{T, T+1, \dots, 2T-1\}$ such that*

$$
D_\Psi(\bar{\mathbf{w}}_T, \mathbf{w}_t) - D_\Psi(\bar{\mathbf{w}}_T, \mathbf{w}_{t+1}) \leq T^{-1} D_\Psi(\bar{\mathbf{w}}_T, \mathbf{w}_T).
$$

*Proof.* We prove this lemma by contradiction. Suppose that

$$
D_\Psi(\bar{\mathbf{w}}_T, \mathbf{w}_t) - D_\Psi(\bar{\mathbf{w}}_T, \mathbf{w}_{t+1}) > T^{-1} D_\Psi(\bar{\mathbf{w}}_T, \mathbf{w}_T), \quad \forall t \in \{T, T+1, \dots, 2T-1\}.
$$

Taking a summation of the above inequality from $t = T$ to $t = 2T - 1$ gives

$$
\begin{aligned}
D_\Psi(\bar{\mathbf{w}}_T, \mathbf{w}_T) - D_\Psi(\bar{\mathbf{w}}_T, \mathbf{w}_{2T}) &= \sum_{t=T}^{2T-1} \big[D_\Psi(\bar{\mathbf{w}}_T, \mathbf{w}_t) - D_\Psi(\bar{\mathbf{w}}_T, \mathbf{w}_{t+1})\big] \\
&> T^{-1} \sum_{t=T}^{2T-1} D_\Psi(\bar{\mathbf{w}}_T, \mathbf{w}_T) = D_\Psi(\bar{\mathbf{w}}_T, \mathbf{w}_T),
\end{aligned}
$$

which contradicts with the non-negativity of the Bregman distance. The proof is complete. $\qquad\square$

# B  Proofs of Convergence Rates for Convex Objectives

In this section, we give proofs of Theorem 1, Theorem 2 and Corollary 3 on the performance of Algorithm 1 for convex objectives.

*Proof of Theorem 1.* Choosing $\mathbf{w} = \mathbf{w}^*$ in (A.1) and using $\sigma_r, \sigma_F \geq 0$, we derive

$$
\mathbb{E}_{z_t}[D_\Psi(\mathbf{w}^*, \mathbf{w}_{t+1})] - D_\Psi(\mathbf{w}^*, \mathbf{w}_t) \leq \eta_t \big[\phi(\mathbf{w}^*) - \phi(\mathbf{w}_t)\big] + \sigma_\Psi^{-1}\eta_t^2 \big[A\phi(\mathbf{w}_t) + 2B\big].
$$

Taking expectations over both sides then gives

$$
\begin{aligned}
\mathbb{E}\big[D_\Psi(\mathbf{w}^*, \mathbf{w}_{t+1}) - D_\Psi(\mathbf{w}^*, \mathbf{w}_t)\big] &\leq \eta_t \mathbb{E}\big[\phi(\mathbf{w}^*) - \phi(\mathbf{w}_t)\big] + \sigma_\Psi^{-1}\eta_t^2 \big[A\mathbb{E}[\phi(\mathbf{w}_t)] + 2B\big] \\
&= \eta_t \mathbb{E}\big[\phi(\mathbf{w}^*) - \phi(\mathbf{w}_t)\big] + \sigma_\Psi^{-1}\eta_t^2 \big[A\mathbb{E}[\phi(\mathbf{w}_t)] - A\phi(\mathbf{w}^*) + A\phi(\mathbf{w}^*) + 2B\big] \\
&= \big(\eta_t - \sigma_\Psi^{-1}\eta_t^2 A\big) \mathbb{E}\big[\phi(\mathbf{w}^*) - \phi(\mathbf{w}_t)\big] + \sigma_\Psi^{-1}\eta_t^2 \big[A\phi(\mathbf{w}^*) + 2B\big] \\
&\leq 2^{-1}\eta_t \mathbb{E}\big[\phi(\mathbf{w}^*) - \phi(\mathbf{w}_t)\big] + \sigma_\Psi^{-1}\eta_t^2 \big[A\phi(\mathbf{w}^*) + 2B\big], \qquad\qquad \text{(B.1)}
\end{aligned}
$$

where we have used the inequality $\eta_t \leq \sigma_\Psi(2A)^{-1}$ and $\phi(\mathbf{w}^*) \leq \phi(\mathbf{w}_t)$. The above inequality can be rewritten as

$$2^{-1}\mathbb{E}\big[\phi(\mathbf{w}_t) - \phi(\mathbf{w}^*)\big] \leq \eta_t^{-1}\mathbb{E}\big[D_\Psi(\mathbf{w}^*, \mathbf{w}_t) - D_\Psi(\mathbf{w}^*, \mathbf{w}_{t+1})\big] + \sigma_\Psi^{-1}\eta_t(A\phi(\mathbf{w}^*) + 2B).$$

Taking a summation of the above inequality from $t = 1$ to $t = T$ gives

$$2^{-1}\sum_{t=1}^{T} \mathbb{E}\big[\phi(\mathbf{w}_t) - \phi(\mathbf{w}^*)\big]$$

$$\leq \sum_{t=1}^{T} \eta_t^{-1}\mathbb{E}\big[D_\Psi(\mathbf{w}^*, \mathbf{w}_t) - D_\Psi(\mathbf{w}^*, \mathbf{w}_{t+1})\big] + \sigma_\Psi^{-1}(A\phi(\mathbf{w}^*) + 2B)\sum_{t=1}^{T} \eta_t$$

$$\leq \sum_{t=2}^{T} \mathbb{E}[D_\Psi(\mathbf{w}^*, \mathbf{w}_t)][\eta_t^{-1} - \eta_{t-1}^{-1}] + \eta_1^{-1}D_\Psi(\mathbf{w}^*, \mathbf{w}_1) + \sigma_\Psi^{-1}(A\phi(\mathbf{w}^*) + 2B)\sum_{t=1}^{T} \eta_t \quad \text{(B.2)}$$

$$\leq D\eta_T^{-1} + 2\sigma_\Psi^{-1}\mu(A\phi(\mathbf{w}^*) + 2B)\sqrt{T},$$

where we have used $\eta_t \leq \eta_{t-1}$ and the elementary inequality $\sum_{t=1}^{T} t^{-\frac{1}{2}} \leq 2\sqrt{T}$ in the last step. Let $\bar{\mathbf{w}}_T = \frac{1}{T}\sum_{t=1}^{T} \mathbf{w}_t$ be defined in Algorithm 1 with $\sigma_\phi = 0$. The convexity of $\phi$ then gives

$$\mathbb{E}[\phi(\bar{\mathbf{w}}_T)] - \phi(\mathbf{w}^*) \leq \frac{2\mu^{-1}D + 4\sigma_\Psi^{-1}\mu(A\phi(\mathbf{w}^*) + 2B)}{\sqrt{T}}. \quad \text{(B.3)}$$

According to Lemma A.2 and the definition of $T^*$ in Algorithm 1, we know

$$D_\Psi(\bar{\mathbf{w}}_T, \mathbf{w}_{T^*}) - D_\Psi(\bar{\mathbf{w}}_T, \mathbf{w}_{T^*+1}) \leq T^{-1}D_\Psi(\bar{\mathbf{w}}_T, \mathbf{w}_T). \quad \text{(B.4)}$$

Choosing $\mathbf{w} = \bar{\mathbf{w}}_T$ in (A.1) followed with expectations over both sides implies ($\bar{\mathbf{w}}_T$ is independent of $z_t$ for any $t \geq T$)

$$\eta_t\mathbb{E}\big[\phi(\mathbf{w}_t) - \phi(\bar{\mathbf{w}}_T)\big] \leq \mathbb{E}[D_\Psi(\bar{\mathbf{w}}_T, \mathbf{w}_t) - D_\Psi(\bar{\mathbf{w}}_T, \mathbf{w}_{t+1})] + \sigma_\Psi^{-1}\eta_t^2\big[A\mathbb{E}[\phi(\mathbf{w}_t)] + 2B\big], \quad \forall t \geq T.$$

Choosing $t = T^*$ in the above inequality and using (B.4) then give

$$\eta_{T^*}\mathbb{E}\big[\phi(\mathbf{w}_{T^*}) - \phi(\bar{\mathbf{w}}_T)\big] \leq \mathbb{E}[D_\Psi(\bar{\mathbf{w}}_T, \mathbf{w}_{T^*}) - D_\Psi(\bar{\mathbf{w}}_T, \mathbf{w}_{T^*+1})] + \sigma_\Psi^{-1}\eta_{T^*}^2\big[A\mathbb{E}[\phi(\mathbf{w}_{T^*})] + 2B\big]$$

$$\leq T^{-1}\mathbb{E}[D_\Psi(\bar{\mathbf{w}}_T, \mathbf{w}_T)] + \sigma_\Psi^{-1}\eta_{T^*}^2\big[A\mathbb{E}[\phi(\mathbf{w}_{T^*})] + 2B\big]. \quad \text{(B.5)}$$

Plugging the step size $\eta_t = \mu/\sqrt{t}$ into the above inequality and using $T \leq T^* \leq 2T$, we derive

$$\mathbb{E}\big[\phi(\mathbf{w}_{T^*}) - \phi(\bar{\mathbf{w}}_T)\big] \leq \frac{\sqrt{2}\mathbb{E}[D_\Psi(\bar{\mathbf{w}}_T, \mathbf{w}_T)]}{\mu\sqrt{T}} + \frac{\mu\big[A\mathbb{E}[\phi(\mathbf{w}_{T^*})] + 2B\big]}{\sigma_\Psi\sqrt{T}},$$

from which and (B.3) it further follows that

$$\mathbb{E}\big[\phi(\mathbf{w}_{T^*}) - \phi(\mathbf{w}^*)\big] = \mathbb{E}\big[\phi(\mathbf{w}_{T^*}) - \phi(\bar{\mathbf{w}}_T)\big] + \mathbb{E}\big[\phi(\bar{\mathbf{w}}_T) - \phi(\mathbf{w}^*)\big] \leq \frac{\sqrt{2}\mathbb{E}[D_\Psi(\bar{\mathbf{w}}_T, \mathbf{w}_T)]}{\mu\sqrt{T}}$$

$$+ \frac{\mu A\big(\mathbb{E}[\phi(\mathbf{w}_{T^*})] - \phi(\mathbf{w}^*)\big)}{\sigma_\Psi\sqrt{T}} + \frac{\mu[A\phi(\mathbf{w}^*) + 2B]}{\sigma_\Psi\sqrt{T}} + \frac{2\mu^{-1}D + 4\sigma_\Psi^{-1}\mu(A\phi(\mathbf{w}^*) + 2B)}{\sqrt{T}} \quad \text{(B.6)}$$

$$\leq \frac{\sqrt{2}D}{\mu\sqrt{T}} + \frac{\mathbb{E}[\phi(\mathbf{w}_{T^*})] - \phi(\mathbf{w}^*)}{2} + \frac{\mu[A\phi(\mathbf{w}^*) + 2B]}{\sigma_\Psi\sqrt{T}} + \frac{2\mu^{-1}D + 4\sigma_\Psi^{-1}\mu(A\phi(\mathbf{w}^*) + 2B)}{\sqrt{T}},$$

where the last inequality is due to $\mu \leq \sigma_\Psi(2A)^{-1}$ and $\phi(\mathbf{w}_{T^*}) \geq \phi(\mathbf{w}^*)$. The above inequality can be written as the stated inequality. The proof is complete. $\quad\square$

*Proof of Theorem 2.* Taking a summation of (B.1) from $t = 1$ to $t = \tilde{t}$, we derive the following inequality for any $\tilde{t} = 0, \ldots, T - 1$

$$\mathbb{E}\big[D_\Psi(\mathbf{w}^*, \mathbf{w}_{\tilde{t}+1})\big] = D_\Psi(\mathbf{w}^*, \mathbf{w}_1) + \sum_{t=1}^{\tilde{t}} \mathbb{E}\big[D_\Psi(\mathbf{w}^*, \mathbf{w}_{t+1}) - D_\Psi(\mathbf{w}^*, \mathbf{w}_t)\big]$$

$$\leq D_\Psi(\mathbf{w}^*, \mathbf{w}_1) + \frac{1}{2}\sum_{t=1}^{\tilde{t}} \eta_t\mathbb{E}\big[\phi(\mathbf{w}^*) - \phi(\mathbf{w}_t)\big] + \sigma_\Psi^{-1}\big[A\phi(\mathbf{w}^*) + 2B\big]\sum_{t=1}^{\tilde{t}} \eta_t^2$$

$$\leq D_{\tilde{t}}, \quad \text{(B.7)}$$

where we have used $\phi(\mathbf{w}^*) \leq \phi(\mathbf{w}_t)$ and $D_{\tilde{t}}$ is defined in Theorem 2. According to the Young's inequality

$$ab \leq \frac{a^s}{s} + \frac{b^{\tilde{s}}}{\tilde{s}} \text{ for all } a, b, s, \tilde{s} > 0 \text{ satisfying } \frac{1}{s} + \frac{1}{\tilde{s}} = 1,$$

the following inequality holds for all $t = 1, 2, \ldots, T$

$$
\begin{aligned}
D_\Psi(\mathbf{w}_t, \mathbf{w}_T) &\leq L_\Psi \|\mathbf{w}_t - \mathbf{w}_T\|^\alpha \leq L_\Psi \left[ 2^{-1}\alpha \|\mathbf{w}_t - \mathbf{w}_T\|^{\alpha \cdot \frac{2}{\alpha}} + 1 - 2^{-1}\alpha \right] \\
&= L_\Psi \left[ 2^{-1}\alpha \|\mathbf{w}_t - \mathbf{w}^* + \mathbf{w}^* - \mathbf{w}_T\|^2 + 1 - 2^{-1}\alpha \right] \\
&\leq L_\Psi \left[ \alpha \|\mathbf{w}_t - \mathbf{w}^*\|^2 + \alpha \|\mathbf{w}^* - \mathbf{w}_T\|^2 + 1 - 2^{-1}\alpha \right] \\
&\leq L_\Psi \left[ 2\sigma_\Psi^{-1}\alpha D_\Psi(\mathbf{w}^*, \mathbf{w}_t) + 2\sigma_\Psi^{-1}\alpha D_\Psi(\mathbf{w}^*, \mathbf{w}_T) + 1 - 2^{-1}\alpha \right],
\end{aligned}
$$

which, coupled with the convexity of $g(\cdot) := D_\Psi(\cdot, \mathbf{w}_T)$ as a function on $\mathbb{R}^d$ and (B.7), then implies

$$\mathbb{E}[D_\Psi(\bar{\mathbf{w}}_T, \mathbf{w}_T)] \leq \frac{1}{T} \sum_{t=1}^T \mathbb{E}[D_\Psi(\mathbf{w}_t, \mathbf{w}_T)] \leq L_\Psi \left[ 4\sigma_\Psi^{-1}\alpha D_T + 1 - 2^{-1}\alpha \right], \tag{B.8}$$

where $\bar{\mathbf{w}}_T$ is defined in Algorithm 1 with $\sigma_\phi = 0$. Analyzing analogously to (B.2) excepting using $\mathbb{E}[D_\Psi(\mathbf{w}^*, \mathbf{w}_t)] \leq D_T$ for all $t \leq T$ ($\eta_t$ is a non-increasing sequence), we derive

$$\mathbb{E}[\phi(\bar{\mathbf{w}}_T)] - \phi(\mathbf{w}^*) \leq \frac{2(A\phi(\mathbf{w}^*) + 2B)\sum_{t=1}^T \eta_t}{T\sigma_\Psi} + \frac{2D_T}{T\eta_T}. \tag{B.9}$$

Eq. (B.5) implies further

$$\mathbb{E}\left[\phi(\mathbf{w}_{T^*}) - \phi(\bar{\mathbf{w}}_T)\right] \leq (T\eta_{T^*})^{-1}\mathbb{E}[D_\Psi(\bar{\mathbf{w}}_T, \mathbf{w}_T)] + \sigma_\Psi^{-1}\eta_{T^*}\left[A\mathbb{E}[\phi(\mathbf{w}_{T^*})] + 2B\right].$$

Combining the above inequality, (B.8), (B.9) together and using $T \leq T^* \leq 2T$, we get

$$
\begin{aligned}
\mathbb{E}\left[\phi(\mathbf{w}_{T^*}) - \phi(\mathbf{w}^*)\right] &= \mathbb{E}\left[\phi(\mathbf{w}_{T^*}) - \phi(\bar{\mathbf{w}}_T)\right] + \mathbb{E}\left[\phi(\bar{\mathbf{w}}_T) - \phi(\mathbf{w}^*)\right] \leq \frac{\mathbb{E}[D_\Psi(\bar{\mathbf{w}}_T, \mathbf{w}_T)]}{T\eta_{2T}} \\
&+ \frac{A\eta_T\left(\mathbb{E}[\phi(\mathbf{w}_{T^*})] - \phi(\mathbf{w}^*)\right)}{\sigma_\Psi} + \frac{\eta_T(A\phi(\mathbf{w}^*) + 2B)}{\sigma_\Psi} + \frac{2(A\phi(\mathbf{w}^*) + 2B)\sum_{t=1}^T \eta_t}{T\sigma_\Psi} + \frac{2D_T}{T\eta_T} \\
&\leq \frac{L_\Psi\left(4\sigma_\Psi^{-1}\alpha D_T + 1\right) + 2D_T}{T\eta_{2T}} + \frac{\mathbb{E}[\phi(\mathbf{w}_{T^*})] - \phi(\mathbf{w}^*)}{2} + \frac{A\phi(\mathbf{w}^*) + 2B}{\sigma_\Psi}\left(\eta_T + \frac{2}{T}\sum_{t=1}^T \eta_t\right),
\end{aligned}
$$

where the last inequality is due to $2A\eta_T \leq \sigma_\Psi$. The above inequality can be written as (3.3). The proof is complete. $\qquad \square$

*Proof of Corollary 3.* (a) Since $\{\eta_t\}$ is a non-increasing sequence, we know

$$0 \leq \lim_{t\to\infty} \eta_t = \lim_{t\to\infty} \frac{t\eta_t^2}{t\eta_t} \leq \lim_{t\to\infty} \frac{1}{t\eta_t} \sum_{\tilde{t}=1}^t \eta_{\tilde{t}}^2 = 0.$$

This in turn shows that the sequence of arithmetic mean also converges to 0, i.e., $\lim_{t\to\infty} \frac{1}{t}\sum_{\tilde{t}=1}^t \eta_{\tilde{t}} = 0$. Also, $\lim_{t\to\infty} \frac{1}{t\eta_t}\sum_{\tilde{t}=1}^t \eta_{\tilde{t}}^2 = 0$ immediately implies $\lim_{t\to\infty} t\eta_t = \infty$. Then, all the terms on the right-hand side of (3.3) converges to zero as $T$ tends to $\infty$ and therefore $\lim_{T\to\infty} \mathbb{E}[\phi(\mathbf{w}_{T^*})] - \phi(\mathbf{w}^*) = 0$.

(b) For the step size sequence $\eta_t = \mu/\sqrt{T}$, (3.3) translates to

$$\mathbb{E}[\phi(\mathbf{w}_{T^*})] - \phi(\mathbf{w}^*) \leq \frac{1}{\sqrt{T}}\left[2L_\Psi\mu^{-1}(4\sigma_\Psi^{-1}\alpha D_T + 1) + 6\mu\sigma_\Psi^{-1}(A\phi(\mathbf{w}^*) + 2B) + 4\mu^{-1}D_T\right],$$

which is of the stated form since in this case $D_T = D_\Psi(\mathbf{w}^*, \mathbf{w}_1) + \sigma_\Psi^{-1}[A\phi(\mathbf{w}^*) + 2B]\mu^2$ is a constant independent of $T$.

The proof is complete. $\qquad \square$

**Remark 1.** If we consider polynomially decaying step sizes $\eta_t = \frac{\mu}{\sqrt{t}}$ for all $t = 1, 2, \ldots, 2T$, then there exists a constant $\widetilde{C} > 0$ independent of $T$ such that

$$\mathbb{E}[\phi(\mathbf{w}_{T^*})] - \phi(\mathbf{w}^*) \leq \frac{\widetilde{C}\log(eT)}{\sqrt{T}}.$$

Indeed, for the step size sequence $\eta_t = \mu/\sqrt{t}$, we have $\sum_{t=1}^T \eta_t \leq 2\mu\sqrt{T}$ and $\sum_{t=1}^T \eta_t^2 \leq \mu^2 \log(eT)$. Plugging these inequalities back into (3.3) gives

$$\mathbb{E}[\phi(\mathbf{w}_{T^*})] - \phi(\mathbf{w}^*) \leq \frac{1}{\sqrt{T}}\Big[2\sqrt{2}L_\Psi\mu^{-1}(4\sigma_\Psi^{-1}\alpha D_T + 1) + 10\mu\sigma_\Psi^{-1}(A\phi(\mathbf{w}^*) + 2B) + 4\sqrt{2}\mu^{-1}D_T\Big],$$

which is of the stated form since in this case $D_T \leq D_\Psi(\mathbf{w}^*, \mathbf{w}_1) + \sigma_\Psi^{-1}[A\phi(\mathbf{w}^*) + 2B]\mu^2 \log(eT)$.

## C   Proofs of Convergence Rates for Strongly Convex Objectives

In this section, we present proofs of convergence rates (Theorem 4, Theorem 5, Theorem 6 and Theorem 7) for strongly convex objectives.

*Proof of Theorem 4.* Choosing $\mathbf{w} = \mathbf{w}^*$ in (A.1) and taking expectation over both sides, we derive the following inequality for any $t \in \mathbb{N}$

$$\begin{aligned}(1 + \sigma_r\eta_t)\mathbb{E}[D_\Psi(\mathbf{w}^*, \mathbf{w}_{t+1})] \leq \\ (1 - \sigma_F\eta_t)\mathbb{E}[D_\Psi(\mathbf{w}^*, \mathbf{w}_t)] + \eta_t\mathbb{E}[\phi(\mathbf{w}^*) - \phi(\mathbf{w}_t)] + \sigma_\Psi^{-1}\eta_t^2[A\mathbb{E}[\phi(\mathbf{w}_t)] + 2B].\end{aligned} \quad \text{(C.1)}$$

For any $t \geq t_0 := \max\{\lceil \frac{8A - 2\sigma_F\sigma_\Psi}{\sigma_\Psi\sigma_\phi}\rceil, 1\}$ ($\lceil a \rceil$ denotes the smallest integer not less than $a$), we have $\eta_t \leq \sigma_\Psi(4A)^{-1}$ and therefore derive

$$\begin{aligned}&(1 + \sigma_r\eta_t)\mathbb{E}[D_\Psi(\mathbf{w}^*, \mathbf{w}_{t+1})] \\ &\leq (1 - \sigma_F\eta_t)\mathbb{E}[D_\Psi(\mathbf{w}^*, \mathbf{w}_t)] + \big(\eta_t - \sigma_\Psi^{-1}\eta_t^2 A\big)\mathbb{E}[\phi(\mathbf{w}^*) - \phi(\mathbf{w}_t)] + \sigma_\Psi^{-1}\eta_t^2[A\phi(\mathbf{w}^*) + 2B] \\ &\leq (1 - \sigma_F\eta_t)\mathbb{E}[D_\Psi(\mathbf{w}^*, \mathbf{w}_t)] + \frac{3\eta_t}{4}\mathbb{E}[\phi(\mathbf{w}^*) - \phi(\mathbf{w}_t)] + \sigma_\Psi^{-1}\eta_t^2[A\phi(\mathbf{w}^*) + 2B], \quad \forall t \geq t_0.\end{aligned}$$

Combining the above inequality and (C.1) together, we derive

$$(1 + \sigma_r\eta_t)\mathbb{E}[D_\Psi(\mathbf{w}^*, \mathbf{w}_{t+1})] \leq (1 - \sigma_F\eta_t)\mathbb{E}[D_\Psi(\mathbf{w}^*, \mathbf{w}_t)] + \frac{3\eta_t}{4}\mathbb{E}[\phi(\mathbf{w}^*) - \phi(\mathbf{w}_t)] + \eta_t^2\widetilde{C}_4 \; \forall t \in \mathbb{N},$$
(C.2)

where

$$\widetilde{C}_4 = \sigma_\Psi^{-1}\big[A\max_{t \leq t_0}\mathbb{E}[\phi(\mathbf{w}_t)] + 2B\big]$$

is a constant independent of $T$. From (C.2), it then follows that

$$\frac{3\eta_t}{4(1 + \sigma_r\eta_t)}\mathbb{E}[\phi(\mathbf{w}_t) - \phi(\mathbf{w}^*)] + \mathbb{E}[D_\Psi(\mathbf{w}^*, \mathbf{w}_{t+1})] \leq \frac{1 - \sigma_F\eta_t}{1 + \sigma_r\eta_t}\mathbb{E}[D_\Psi(\mathbf{w}^*, \mathbf{w}_t)] + \eta_t^2\widetilde{C}_4, \; \forall t \in \mathbb{N}.$$
(C.3)

Since $\eta_t = \frac{2}{\sigma_\phi t + 2\sigma_F}$, we know

$$\frac{1 - \sigma_F\eta_t}{1 + \sigma_r\eta_t} = \frac{\sigma_\phi t + 2\sigma_F - 2\sigma_F}{\sigma_\phi t + 2\sigma_F + 2\sigma_r} = \frac{t}{t + 2}$$

and

$$\frac{1}{1 + \sigma_r\eta_t} \geq \frac{1}{1 + \sigma_r\eta_1} = \frac{\sigma_\phi + 2\sigma_F}{3\sigma_\phi} \geq \frac{1}{3}.$$

Plugging the above two inequalities into Eq. (C.3), we derive

$$4^{-1}\eta_t\mathbb{E}[\phi(\mathbf{w}_t) - \phi(\mathbf{w}^*)] + \mathbb{E}[D_\Psi(\mathbf{w}^*, \mathbf{w}_{t+1})] \leq \frac{t}{t + 2}\mathbb{E}[D_\Psi(\mathbf{w}^*, \mathbf{w}_t)] + \eta_t^2\widetilde{C}_4, \quad \forall t \in \mathbb{N}.$$

Multiplying both sides by $(t+1)(t+2)$ gives

$$4^{-1}(t+1)(t+2)\eta_t\mathbb{E}[\phi(\mathbf{w}_t) - \phi(\mathbf{w}^*)] + (t+1)(t+2)\mathbb{E}[D_\Psi(\mathbf{w}^*, \mathbf{w}_{t+1})]$$
$$\leq t(t+1)\mathbb{E}[D_\Psi(\mathbf{w}^*, \mathbf{w}_t)] + (t+1)(t+2)\eta_t^2\widetilde{C}_4, \ \forall t \in \mathbb{N}.$$

Taking a summation of the above inequality from $t = 1$ to $t = T$ gives

$$4^{-1}\sum_{t=1}^{T}\Big[(t+1)(t+2)\eta_t\mathbb{E}[\phi(\mathbf{w}_t) - \phi(\mathbf{w}^*)]\Big] + (T+1)(T+2)\mathbb{E}[D_\Psi(\mathbf{w}^*, \mathbf{w}_{T+1})]$$

$$\leq 2D_\Psi(\mathbf{w}^*, \mathbf{w}_1) + \widetilde{C}_4\sum_{t=1}^{T}(t+1)(t+2)\eta_t^2 \leq 2D_\Psi(\mathbf{w}^*, \mathbf{w}_1) + \frac{4\widetilde{C}_4}{\sigma_\phi^2}\sum_{t=1}^{T}\frac{(t+1)(t+2)}{t^2}$$

$$\leq 2D_\Psi(\mathbf{w}^*, \mathbf{w}_1) + \frac{4\widetilde{C}_4}{\sigma_\phi^2}\big[T + 3\log(eT) + 4\big]$$

$$\leq \sigma_\phi^{-2}\big[4\widetilde{C}_4(T + 3e^{-1}T + 7) + 2D_\Psi(\mathbf{w}^*, \mathbf{w}_1)\sigma_\phi^2\big] \leq \widetilde{C}_2\sigma_\phi^{-2}T,$$

where we have used the elementary inequality $\log(T) \leq e^{-1}T$ for any $T \in \mathbb{N}$ and introduced

$$\widetilde{C}_2 = 4\widetilde{C}_4(3e^{-1} + 8) + 2D_\Psi(\mathbf{w}^*, \mathbf{w}_1)\sigma_\phi^2.$$

It then follows that

$$\mathbb{E}[D_\Psi(\mathbf{w}^*, \mathbf{w}_{T+1})] \leq \widetilde{C}_2\sigma_\phi^{-2}(T+2)^{-1}$$

and

$$\mathbb{E}[\phi(\bar{\mathbf{w}}_T) - \phi(\mathbf{w}^*)] \leq \frac{\sum_{t=1}^{T}(t+1)(t+2)\eta_t\mathbb{E}[\phi(\mathbf{w}_t) - \phi(\mathbf{w}^*)]}{\sum_{t=1}^{T}(t+1)(t+2)\eta_t}$$

$$\leq \frac{4\widetilde{C}_2 T}{\sigma_\phi^2\sum_{t=1}^{T}(t+1)(t+2)\eta_t} \leq \frac{4\widetilde{C}_2}{(T+1)\sigma_\phi}.$$

Here we have used the inequality

$$\sum_{t=1}^{T}(t+1)(t+2)\eta_t \geq \sum_{t=1}^{T}\frac{2(t+1)(t+2)}{\sigma_\phi t} \geq \frac{2}{\sigma_\phi}\sum_{t=1}^{T}t = \frac{T(T+1)}{\sigma_\phi}.$$

The proof is complete. $\qquad\qquad\qquad\qquad\qquad\qquad\qquad\qquad\qquad\qquad\qquad\qquad\square$

We will use the following lemma to prove sufficient conditions for the convergence of SCMD established in Theorem 5. The following lemma is known in the literature (see, e.g., [3, 5]).

**Lemma C.1.** *Let* $\{\eta_t\}_{t\in\mathbb{N}}$ *be a sequence of non-negative numbers such that* $\lim_{t\to\infty}\eta_t = 0$ *and* $\sum_{t=1}^{\infty}\eta_t = \infty$. *Let* $a > 0$ *and* $t_1 \in \mathbb{N}$ *such that* $\eta_t < a^{-1}$ *for any* $t \geq t_1$. *Then we have*

$$\lim_{T\to\infty}\sum_{t=t_1}^{T}\eta_t^2\prod_{k=t+1}^{T}(1 - a\eta_k) = 0.$$

*Proof of Theorem 5.* Since $\lim_{t\to\infty} = 0$, there exists a $\tilde{t}_0 \in \mathbb{N}$ such that $\eta_t \leq \sigma_\Psi(4A)^{-1}$ and $\eta_t \leq \sigma_r^{-1}$ for all $t \geq \tilde{t}_0$. From (C.2) and $\eta_t\sigma_r \leq 1$ it follows that

$$\mathbb{E}[D_\Psi(\mathbf{w}^*, \mathbf{w}_{t+1})] \leq \frac{1 + \sigma_r\eta_t - \sigma_\phi\eta_t}{1 + \sigma_r\eta_t}\mathbb{E}[D_\Psi(\mathbf{w}^*, \mathbf{w}_t)] + \eta_t^2\widetilde{C}_5$$

$$\leq \big(1 - 2^{-1}\sigma_\phi\eta_t\big)\mathbb{E}[D_\Psi(\mathbf{w}^*, \mathbf{w}_t)] + \eta_t^2\widetilde{C}_5, \quad \forall t \geq \tilde{t}_0,$$

where

$$\widetilde{C}_5 = \sigma_\Psi^{-1}\big[A\max_{t\leq\tilde{t}_0}\mathbb{E}[\phi(\mathbf{w}_t)] + 2B\big].$$

Applying this inequality iteratively for $t = T, \ldots, \tilde{t}_0$ yields

$$\mathbb{E}[D_\Psi(\mathbf{w}^*, \mathbf{w}_{T+1})] \le \prod_{t=\tilde{t}_0}^{T}(1-2^{-1}\sigma_\phi\eta_t)\mathbb{E}[D_\Psi(\mathbf{w}^*, \mathbf{w}_{\tilde{t}_0})] + \widetilde{C}_5 \sum_{t=\tilde{t}_0}^{T}\eta_t^2 \prod_{k=t+1}^{T}(1-2^{-1}\sigma_\phi\eta_k), \quad \text{(C.4)}$$

where we denote $\prod_{k=t+1}^{T}(1 - 2^{-1}\sigma_\phi\eta_k) = 1$ for $t = T$. The first term of the above inequality can be controlled by the standard inequality $1 - a \le \exp(-a), a > 0$ together with $\sum_{t=1}^{\infty}\eta_t = \infty$

$$\lim_{T\to\infty} \prod_{t=\tilde{t}_0}^{T}(1 - 2^{-1}\sigma_\phi\eta_t)\mathbb{E}[D_\Psi(\mathbf{w}^*, \mathbf{w}_{\tilde{t}_0})] \le \lim_{T\to\infty}\prod_{t=\tilde{t}_0}^{T}\exp\big(-2^{-1}\sigma_\phi\eta_t\big)\mathbb{E}[D_\Psi(\mathbf{w}^*, \mathbf{w}_{\tilde{t}_0})]$$

$$= \lim_{T\to\infty}\exp\Big(-2^{-1}\sigma_\phi\sum_{t=\tilde{t}_0}^{T}\eta_t\Big)\mathbb{E}[D_\Psi(\mathbf{w}^*, \mathbf{w}_{\tilde{t}_0})] = 0.$$

Applying Lemma C.1 with $a = 2^{-1}\sigma_\phi$, we get

$$\lim_{T\to\infty}\sum_{t=\tilde{t}_0}^{T}\eta_t^2\prod_{k=t+1}^{T}(1-2^{-1}\sigma_\phi\eta_k) = 0.$$

Plugging the above two expressions into (C.4) completes the proof. $\qquad\square$

*Proof of Theorem 6.* According to the strong convexity of $\phi$ given in (3.2) and the optimality condition $0 \in \partial\phi(\mathbf{w}^*)$, we get

$$\mathbb{E}[\phi(\bar{\mathbf{w}}_T) - \phi(\mathbf{w}^*)] = \mathbb{E}[\phi(\bar{\mathbf{w}}_T) - \phi(\mathbf{w}^*) - \langle\bar{\mathbf{w}}_T - \mathbf{w}^*, 0\rangle] \ge \sigma_\phi\mathbb{E}[D_\Psi(\bar{\mathbf{w}}_T, \mathbf{w}^*)],$$

which, together with the strong convexity of $\Psi$ and the first inequality of (3.4), gives

$$\mathbb{E}[\|\bar{\mathbf{w}}_T - \mathbf{w}^*\|^2] \le \frac{2\mathbb{E}[\phi(\bar{\mathbf{w}}_T) - \phi(\mathbf{w}^*)]}{\sigma_\phi\sigma_\Psi} \le \frac{8\widetilde{C}_2}{(T+1)\sigma_\phi^2\sigma_\Psi}.$$

It then follows from the $L_\Psi$-strong smoothness of $\Psi$, $\mathbb{E}[\|\mathbf{w}^* - \mathbf{w}_T\|^2] \le \frac{2}{\sigma_\Psi}\mathbb{E}[D_\Psi(\mathbf{w}^*, \mathbf{w}_T)]$ and the second inequality of (3.4) that

$$\mathbb{E}[D_\Psi(\bar{\mathbf{w}}_T, \mathbf{w}_T)] \le 2^{-1}L_\Psi\mathbb{E}[\|\bar{\mathbf{w}}_T - \mathbf{w}_T\|^2] = 2^{-1}L_\Psi\mathbb{E}\big[\|\bar{\mathbf{w}}_T - \mathbf{w}^* + \mathbf{w}^* - \mathbf{w}_T\|^2\big]$$

$$\le L_\Psi\mathbb{E}\big[\|\bar{\mathbf{w}}_T - \mathbf{w}^*\|^2 + \|\mathbf{w}^* - \mathbf{w}_T\|^2\big] \le \frac{10L_\Psi\widetilde{C}_2}{(T+1)\sigma_\phi^2\sigma_\Psi}.$$

Plugging the above inequality back into (B.5) and using $T \le T^* \le 2T - 1$ give

$$\mathbb{E}\big[\phi(\mathbf{w}_{T^*}) - \phi(\bar{\mathbf{w}}_T)\big] \le T^{-1}\eta_{T^*}^{-1}\mathbb{E}[D_\Psi(\bar{\mathbf{w}}_T, \mathbf{w}_T)] + \sigma_\Psi^{-1}\eta_{T^*}\big[A\mathbb{E}[\phi(\mathbf{w}_{T^*})] + 2B\big]$$

$$\le \frac{\sigma_\phi(T^* + 2)}{2T}\frac{10L_\Psi\widetilde{C}_2}{(T+1)\sigma_\phi^2\sigma_\Psi} + \frac{2\big[A\mathbb{E}[\phi(\mathbf{w}_{T^*})] + 2B\big]}{\sigma_\Psi\sigma_\phi T^*}$$

$$= \frac{10L_\Psi\widetilde{C}_2}{T\sigma_\phi\sigma_\Psi} + \frac{2\big[A\mathbb{E}[\phi(\mathbf{w}_{T^*})] + 2B\big]}{\sigma_\Psi\sigma_\phi T}. \qquad \text{(C.5)}$$

Plugging the first inequality in (3.4) and (C.5) back into the error decomposition (2.5) gives

$$\mathbb{E}\big[\phi(\mathbf{w}_{T^*}) - \phi(\mathbf{w}^*)\big] \le \frac{10L_\Psi\widetilde{C}_2\sigma_\Psi^{-1} + 4\widetilde{C}_2}{T\sigma_\phi} + \frac{2A\big[\mathbb{E}[\phi(\mathbf{w}_{T^*})] - \phi(\mathbf{w}^*)\big]}{\sigma_\Psi\sigma_\phi T} + \frac{2\big[A\phi(\mathbf{w}^*) + 2B\big]}{\sigma_\Psi\sigma_\phi T}$$

$$\le \frac{10L_\Psi\widetilde{C}_2\sigma_\Psi^{-1} + 4\widetilde{C}_2}{T\sigma_\phi} + \frac{\mathbb{E}[\phi(\mathbf{w}_{T^*})] - \phi(\mathbf{w}^*)}{2} + \frac{2\big[A\phi(\mathbf{w}^*) + 2B\big]}{\sigma_\Psi\sigma_\phi T},$$

where we have used the inequality $\frac{2A}{\sigma_\Psi\sigma_\phi T} \le \frac{1}{2}$ and $\phi(\mathbf{w}_{T^*}) - \phi(\mathbf{w}^*) \ge 0$ in the last step. The above inequality can be written as stated inequality with

$$\widetilde{C}_3 = 4\widetilde{C}_2(5L_\Psi\sigma_\Psi^{-1} + 2) + 4\sigma_\Psi^{-1}\big[A\phi(\mathbf{w}^*) + 2B\big].$$

The proof is complete. $\qquad\square$

Finally, we give the proof of Theorem 7 on lower bounds of convergence rates under a lower-bound assumption on the variance of $\nabla f(\mathbf{w}, z)$ as an unbiased estimate of $\nabla F(\mathbf{w})$.

*Proof of Theorem 7.* If $\Psi(\mathbf{w}) = \frac{1}{2}\|\mathbf{w}\|_2^2$ and $r(\mathbf{w}) = 0$, then (2.2) becomes

$$\mathbf{w}_{t+1} = \mathbf{w}_t - \eta_t \nabla f(\mathbf{w}_t, z_t),$$

from which we know

$$\|\mathbf{w}_{t+1} - \mathbf{w}^*\|_2^2 = \|\mathbf{w}_t - \mathbf{w}^*\|_2^2 + \eta_t^2 \|\nabla f(\mathbf{w}_t, z_t)\|_2^2 - 2\eta_t \langle \mathbf{w}_t - \mathbf{w}^*, \nabla f(\mathbf{w}_t, z_t) \rangle.$$

Taking expectations over both sides, we get

$$\mathbb{E}[\|\mathbf{w}_{t+1} - \mathbf{w}^*\|_2^2] = \mathbb{E}[\|\mathbf{w}_t - \mathbf{w}^*\|_2^2] + \eta_t^2 \mathbb{E}[\|\nabla f(\mathbf{w}_t, z_t)\|_2^2] - 2\eta_t \mathbb{E}[\langle \mathbf{w}_t - \mathbf{w}^*, \nabla \phi(\mathbf{w}_t) \rangle]. \quad \text{(C.6)}$$

According to the lower bound assumption on variances, we know

$$\mathbb{E}_{z_t}[\|\nabla f(\mathbf{w}_t, z_t)\|_2^2] = \mathbb{E}_{z_t}[\|\nabla f(\mathbf{w}_t, z_t) - \nabla F(\mathbf{w}_t)\|_2^2] + \|\nabla F(\mathbf{w}_t)\|_2^2 \geq \sigma^2.$$

We can combine the above inequality and (C.6) to derive

$$\mathbb{E}[\|\mathbf{w}_{t+1} - \mathbf{w}^*\|_2^2] \geq \mathbb{E}[\|\mathbf{w}_t - \mathbf{w}^*\|_2^2] + \eta_t^2 \sigma^2 - 2\eta_t \mathbb{E}[\langle \mathbf{w}_t - \mathbf{w}^*, \nabla \phi(\mathbf{w}_t) \rangle]. \quad \text{(C.7)}$$

Since $\phi$ is $L_\phi$-smooth, we know

$$\langle \mathbf{w}^* - \mathbf{w}_t, \nabla \phi(\mathbf{w}_t) \rangle \geq -L_\phi \|\mathbf{w}^* - \mathbf{w}_t\|_2^2,$$

which, plugged into (C.6) with the assumption $2L_\phi \eta_t \leq 1$, implies the following inequality

$$\mathbb{E}[\|\mathbf{w}_{t+1} - \mathbf{w}^*\|_2^2] \geq (1 - 2L_\phi \eta_t)\mathbb{E}[\|\mathbf{w}_t - \mathbf{w}^*\|_2^2] + 2L_\phi \eta_t (\eta_t \sigma^2/(2L_\phi)),$$

from which we know

$$\mathbb{E}[\|\mathbf{w}_{t+1} - \mathbf{w}^*\|_2^2] \geq \min\{\mathbb{E}\|\mathbf{w}_t - \mathbf{w}^*\|_2^2, \eta_t \sigma^2/(2L_\phi)\}.$$

We can apply the above inequality iteratively to show

$$\mathbb{E}[\|\mathbf{w}_{t+1} - \mathbf{w}^*\|_2^2] \geq \min\{\|\mathbf{w}_1 - \mathbf{w}^*\|_2^2, \eta_1 \sigma^2/(2L_\phi), \ldots, \eta_t \sigma^2/(2L_\phi)\}.$$

The proof is complete. $\square$

Table C.1: Description of the datasets used in the experiments.

| datasets | # inst | # feat | datasets | # inst | # feat | datasets | # inst | # feat | datasets | # inst | # feat |
|---|---|---|---|---|---|---|---|---|---|---|---|
| diabetes | 768 | 8 | german | 1000 | 24 | splice | 1000 | 60 | usps | 7291 | 256 |
| mnist | 60000 | 780 | w8a | 49749 | 300 | letter | 15000 | 16 | satimage | 4456 | 36 |
| ijcnn1 | 141691 | 22 | mushrooms | 8124 | 112 | a9a | 32561 | 123 | connect | 67557 | 126 |
| cover | 286048 | 10 | webspam_u | 350000 | 254 | real-sim | 72309 | 20958 | rcv1 | 20242 | 47236 |

# D Additional Experimental Results

In this section, we give the description of datasets used in the experiments and report more experimental results.

## D.1 Description of Datasets

In this subsection, we provide in Table C.1 the information for the datasets used in Section 5.1. Webspam_u is a subset used in the Pascal Large Scale Learning Challenge [4] to detect malicious web pages. The remaining datasets can be downloaded from the LIBSVM homepage [2].

## D.2 Testing errors versus iteration numbers

In this subsection, we compare the behavior of several variants of SGD and SPGD on testing datasets. In Figure D.1, we plot the objective function values on testing datasets versus iteration numbers for SPGD (mean$\pm$0.5std). In Figure D.2, we plot the objective function values on testing datasets versus iteration numbers for SGD applied to different datasets.

Figure D.1: Objective function values on test datasets versus iteration numbers for SPGD.

### D.3 Training errors versus iteration numbers

In this section, we report more experimental results on training errors. In Figure D.3 and Figure D.4, we report the objective function values on training examples versus the number of iterations for SPGD and SGD, which behave analogously to testing errors.

### D.4 Additional experimental results for tomography reconstruction

In this section, we report more experimental results for tomography reconstruction. In Figure D.5, we compare the behavior of our method with several baseline methods for the tomography reconstruction problem with $N = 32, n = 11520$ and $5\%$ relative noise.

Figure D.2: Objective function values on test datasets versus iteration numbers for SGD.

(a) diabetes    (b) german    (c) splice    (d) usps

(e) mnist    (f) w8a    (g) letter    (h) satimage

(i) ijcnn1    (j) mushroom    (k) a9a    (l) connect

(m) cover    (n) webspam_u    (o) real-sim    (p) rcv1

Figure D.3: Objective function values on training datasets versus iteration numbers for SPGD.

(a) diabetes     (b) german     (c) splice     (d) usps

(e) mnist     (f) w8a     (g) letter     (h) satimage

(i) ijcnn1     (j) mushroom     (k) a9a     (l) connect

(m) cover     (n) webspam_u     (o) real-sim     (p) rcv1

Figure D.4: Objective function values on training datasets versus iteration numbers for SGD.

(a) True Image.     (b) Reconstructed Image.     (c) Error vs. $T$.     (d) NNCs vs. $T$

Figure D.5: Tomography reconstruction with $N = 32, n = 11520$ and $5\%$ relative noise. Panel (a) and (b) are the true image and the reconstructed image by OCMDI, respectively. Panel (c) and (d) plot the errors and NNCs versus iteration numbers.