[Reviews · NeurIPS 2019]

Reviewer 1



originality, significance: I am not sufficiently familiar with the contextual literature to deeply judge these aspects. However the ability to preserve sparsity over similar methods with such convergence guarantees is significant. quality: The quality seems high, but I wonder about the lack of error bars. How does the method react to random initializations? The large number of datasets makes me not worry too much about that-- but it seems like you did more than sufficient number of experiments to generate such uncertainty estimates so why aren't they shown? clarity: The writing is very clear and a pleasure to read. ==================================== ==================================== Thank you for your thorough response and for including the error bars.

Reviewer 2



*the paper is very well written; main ideas are explained clearly; results are presented mathematically rigorously *I have only a view comments which refer to comparing the proposed strategy to taking just the last iterate: -Algorithm 1: when I understand correctly, one has to calculate all iterates up to t=2T-1 and needs to store all iterates from t=T up to t=2T-1 -is the map t --> A_t monotone under the assumption of (strong) convexity ? this question refers to the choice of T* in Algorithm 1 - wouldn't any t \geq T* also do the job? ... the best choice would possibly be the argmin of all t satisfying the condition in l.16 (which is possibly the last iterate) -could you comment on this? *the benefit of this strategy compared to just taking some kind of averaging seems clear; but I do not see the benefit compared to taking the last iterate

Reviewer 3



The paper makes an important contribution to the literature of SGD optimization algorithms in that it introduces an algorithm that achieves (1) last round convergence, (2) allows for sparsity constraints imposed by regularizer, and (3) achieves the optimal convergence rate. The algorithm is a simple variant of the SCMD algorithm. The original version either gets (1) and (2) with suboptimal (up to a log T factor) convergence rate, or with time-averaging achieves (3). The main change in this paper is a subroutine that searches for a specific round that admits the optimal convergence rate, the existence of such a round is guaranteed by the fact that time-averaging gets optimal convergences; searching for such a round, however, is non-trivial. Experiments on real-world dataset clearly show the competitive edge of the new algorithm.

[Author Response · NeurIPS 2019]

We appreciate the reviewers for the time and expertise they have invested in writing these constructive comments.

**Reviewer #1**

**Q**: *The lack of error bars. How does the method react to random initializations? Why aren't uncertainty shown?*

**A**: Thank you for your constructive suggestion, according to which we draw the error bars (mean $\pm$ std) to show how
the method reacts to random initializations. Please see Panel (a) of Figure I for an example. We will use error bars to
present our experimental results in the camera-ready version.

**Q**: *To increase my score even higher I need to be convinced that the theoretical result is a very substantial advance.*

**A**: Thanks. The significance of our theoretical contribution is to find a simple strategy to identify a single iterate from
the iterate sequence with optimal convergence rates. While the existence of such an iterate is guaranteed by the fact that
time-averaging gets optimal convergence, searching for such an iterate is non-trivial. Our method also has a potential to
be applicable to other stochastic algorithms, e.g., stochastic dual averaging.

**Reviewer #2**

**Q**: *Algorithm 1 (Alg. 1): when I understand correctly, one has to calculate all iterates up to $t = 2T - 1$ and needs to*
*store all iterates from $t = T$ up to $t = 2T - 1$.*

**A**: Thanks for the careful observation. Our description of Alg. 1 leaves an impression that it needs to store all iterates
from $t = T$ up to $t = 2T - 1$ since we set $T^*$ in line 17 of Alg. 1. However, this storage is indeed not required if we set
$\mathbf{w}_{T^*} \leftarrow \mathbf{w}_t$ in line 17 of Alg. 1 (we only need $\mathbf{w}_{T^*}$ in practical implementation). We will address this in the revision.

**Q**: *Is the map $t \mapsto A_t$ monotone under (strong) convexity assumption? this refers to the choice of $T^*$ in Algorithm 1*

**A**: Thanks for the query. Motivated by your comment, we run an experiment on SVM problems with a strongly convex
objective to check the monotonicity of $A_t$. In Panel (b) of Figure I, we plot $A_t$ as a function of $t$, from which we see
that $A_t$ is not a monotone function of $t$. We will mention it in the camera-ready version.

**Q**: *Wouldn't any $t \geq T^*$ also do the job? ... the best choice would possibly be the $\arg\min$ of all $t$ satisfying the*
*condition in l.16 (which is possibly the last iterate)*

**A**: Thanks for the query. We conjecture that not all $t \geq T^*$ can achieve optimal convergence. The underlying reason is
that $t \geq T^*$ may not necessarily satisfy the condition in line 16 of Alg. 1, which is required to get optimal convergence
in our analysis.

Among all $t$ satisfying the condition in line 16 of Alg. 1, the minimal $t$ (MIN-T) has an appealing property of requiring
the minimal computational cost, whose performance may be further improved if we update the model once encountering
an $\mathbf{w}_{t'}$ satisfying the condition in line 16 of Alg. 1 with $t' > t$. The intuition is that the added computational cost may
generally come along with a better model. This is the strategy adopted by SCMDI/OCMDI. Another strategy is to set
$T^*$ as the index whose associated $\triangle$ is minimal (MIN-A). The intuition is that the quality of $\mathbf{w}_t$ depends on $\triangle$ (please
see line 243 of the paper). We run an experiment to show how OCMDI behaves versus MIN-T and MIN-A, and report
results in Panel (c) of Figure I. We will add a comment in the camera-ready version.

**Q**: *Benefit compared to just taking averaging seems clear; I do not see the benefit compared to taking the last iterate*

**A**: Thank you for the comment. The benefit compared to taking the last iterate mainly consists in the theoretical property.
Taking the last iterate can only achieve a suboptimal convergence rate with high probabilities (up to a $\log T$ factor),
while our strategy can achieve the optimal convergence rate.

**Reviewer #3**: Thank you for your very positive comments.

(a) Error Bar Presentation     (b) Monotonicity     (c) Comparison on Selecting Iterates

Figure I: Experimental results of SPGD applied to SVM problems with the data Splice.

[Meta-Review · NeurIPS 2019]

This paper proposes an online stochastic optimization algorithm (similar to SGD) that has optimal convergence rate of the last iterate in two settings (O(1/sqrt(T)) for Lipschitz convex functions and O(1/T) strongly convex functions), and additionally it allows an arbitrary non-smooth regularizer (e.g. L1-norm to induce sparsity). Many subsets of the properties are achieved by prior works. Namely, it was known how to achieve these results up to O(log T) factors. It was known how to achieve the optimal rates with averaging, which, however, destroys sparsity. However, this paper has the first algorithm that has all the properties simultaneously and removes the log factors. The paper has rigorous proofs of the convergence rates and extensive numerical experiments. While many mathematical techniques used in the convergence analysis are standard, several key steps in the algorithm and the analysis are non-trivial, which make this paper an important contribution to stochastic optimization research. The paper is clearly written, the proofs appear to be correct. The authors addressed all the suggestions and questions from the reviewers in their rebuttal.